# IndicFake Meets SAFARI-LLM: Unifying Semantic and Acoustic Intelligence for Multilingual Deepfake Detection

**Rishabh Ranjan**                                                                  *ranjan.4@iitj.ac.in*
*Department of Computer Science*
*Indian Institute of Technology Jodhpur*

**Mayank Vatsa**                                                                    *mvatsa@iitj.ac.in*
*Indian Institute of Technology Jodhpur*

**Richa Singh**                                                                     *richa@iitj.ac.in*
*Indian Institute of Technology Jodhpur*

**Reviewed on OpenReview:** *https://openreview.net/forum?id=s8pPYRVVTU*

## Abstract

Audio deepfakes pose a growing threat, particularly in linguistically diverse and low-resource settings where existing detection methods often struggle. This work introduces two transformative contributions to address these challenges. First, we present **IndicFake**, a pioneering audio deepfake dataset with over 4.2 million samples (7,350 hours) spanning English and 17 Indian languages across Indo-European, Dravidian, and Sino-Tibetan families. With minimal overlap (Jaccard similarity: 0.00–0.06) with existing datasets, IndicFake offers an unparalleled benchmark for multilingual deepfake detection. Second, we propose **SAFARI-LLM** (Semantic Acoustic Feature Adaptive Router with Integrated LLM). This novel framework integrates Whisper's semantic embeddings and m-HuBERT's acoustic features through an adaptive Audio Feature Unification Module (AFUM). Enhanced by LoRA-fine-tuned LLaMA-7B, SAFARI-LLM achieves unmatched cross-lingual and cross-family generalization. Evaluations across the IndicFake, DECRO, and WaveFake datasets demonstrate its superiority, outperforming 14 state-of-the-art models with standout accuracies of 94.21% (English-to-Japanese transfer on WaveFake) and 84.48% (English-to-Chinese transfer on DECRO), alongside robust performance across diverse linguistic contexts. These advancements establish a new standard for reliable, scalable audio deepfake detection. Resources will be publicly available at: URL.

## 1 Introduction

Voice technology has fundamentally changed how we engage with devices and services. Driven by sophisticated speech recognition that converts spoken words into text with remarkable precision, it powers assistants such as Siri, Alexa, and Google Assistant, enabling a wide range of tasks through simple voice commands. In the United States, approximately 128 million people[1] use these assistants each month. Meanwhile, smart speaker adoption has increased by 135% since 2018, highlighting the rapid growth of voice-focused technology. Beyond convenience, voice interfaces are becoming increasingly vital for security through speaker verification, utilizing unique vocal features as biometric markers. Financial institutions such as Wells Fargo and Barclays[2] now utilize this technology, enabling secure account management through voice commands. Indian Railways has similarly introduced voice-enabled ticket booking, streamlining travel for millions.

---

[1]http://tinyurl.com/usvoice135
[2]http://tinyurl.com/barclaysvoice

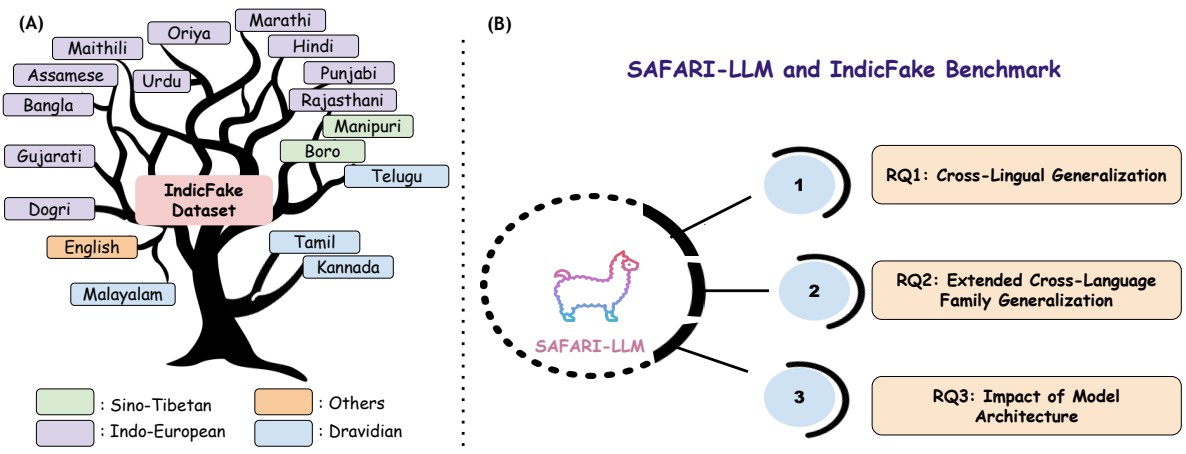

Figure 1: Overview of the IndicFake benchmark. SAFARI-LLM integrates language-specific and universal features for robust detection, addressing linguistic diversity. The IndicFake dataset spans 18 languages across Indo-European, Dravidian, Sino-Tibetan, and other language families, allowing for a comprehensive evaluation. SAFARI-LLM integrates language-specific and universal features for robust detection, addressing linguistic diversity.

These advancements also present significant risks. A recent report[3] indicated nearly 50 million AI-generated "voice clone" calls spanning 22 official languages during the two months preceding India's General Elections, highlighting the growing threat of deepfakes exploiting linguistic diversity. Such deepfakes pose severe implications, including the potential for election manipulation, financial fraud, and social engineering attacks. Moreover, the ability of these deepfake technologies to convincingly replicate individual voices, including those of public figures, adds layers of complexity to authentication and verification processes, necessitating urgent advancements in detection technologies. Existing studies explore various deepfake algorithms, but real-world complexities remain underexamined, notably the impact of accent variations, dialectal differences, and the robustness of models across linguistically diverse settings (Ranjan et al., 2023; 2024).

Furthermore, voice-based systems are particularly vulnerable in multilingual contexts, where training data predominantly consists of high-resource languages, resulting in less effective coverage of low-resource languages. Such discrepancies in data availability significantly impact the detection efficacy, highlighting an imbalance that undermines the overall security robustness. These gaps emphasize the urgent need for extensive and representative datasets to facilitate practical training and evaluation of deepfake detection systems in multilingual and cross-linguistic scenarios.

## 1.1 Related Work

Over the past decade, numerous voice anti-spoofing datasets have been developed to address deepfake audio challenges. Benchmark datasets, such as ASVspoof (Wu et al., 2015; Wang et al., 2020; Yamagishi et al., 2021; Wang et al., 2024) and Audio Deepfake Detection (ADD) (Yi et al., 2022; 2023), have spearheaded tasks like fake audio detection and manipulation region localization. However, these datasets generally feature English samples with limited noise and codec variations (Reimao & Tzerpos, 2019; Frank & Schönherr, 2021; Ma et al., 2022), reducing their effectiveness in diverse acoustic scenarios. Datasets such as MLAAD (Müller et al., 2024) and DECRO (Ba et al., 2023) expand linguistic coverage but lack partial fakes (Yi et al., 2022). Efforts like HABLA (Tamayo Flórez et al., 2023), ILLUSION (Thakral et al., 2025), CVoiceFake (Li et al., 2024), and VoiceWukong (Yan et al., 2024) broaden non-English contexts. Yet, no dataset comprehensively offers extensive multilingual coverage, advanced generation methods, and real-world variability (Table 1).

Algorithmically, early approaches employed handcrafted features, such as phase (Xiao et al., 2015), magnitude (Tian et al., 2016), and pitch (Korshunov & Marcel, 2016), or deep learning models using raw waveforms

---

[3]https://tinyurl.com/indianelectionas

Table 1: Summarizing the key statistics of the proposed IndicFake dataset and its comparison with existing speech deepfake datasets. IndicFake is the largest among all the existing datasets and covers 18 languages of the Southeast Asian region.

| Dataset | Year | Language | Indic Languages | Spoofed Methods | # Total Samples |
|---|---|---|---|---|---|
| ASVspoof 2015 (Wu et al., 2015) | 2015 | English | ✗ | 10 | 246,500 |
| ASVspoof 2019-LA (Wang et al., 2020) | 2019 | English | ✗ | 19 | 130,378 |
| FoR (Reimao & Tzerpos, 2019) | 2019 | English | ✗ | 7 | 87,285 |
| ASVspoof 2021-LA (Yamagishi et al., 2021) | 2021 | English | ✗ | 19 | 148,148 |
| ASVspoof 2021-DF (Yamagishi et al., 2021) | 2021 | English | ✗ | 100+ | 572,616 |
| WaveFake (Frank & Schönherr, 2021) | 2021 | English, Japanese | ✗ | 7 | 117,985 |
| ADD2022 -LF (Yi et al., 2022) | 2022 | Chinese | ✗ | Unknown | 389,419 |
| Latin American (Tamayo Flórez et al., 2023) | 2022 | Spanish | ✗ | 6 | 58,000 |
| CFAD (Ma et al., 2022) | 2023 | Chinese | ✗ | 12 | 231,600 |
| DECRO (Ba et al., 2023) | 2024 | English, Chinese | ✗ | 10 | 118,381 |
| MLAAD (Müller et al., 2024) | 2024 | 38 Languages | ✓(Hindi, Bangla) | 26 | 82,000 |
| ASVspoof5 (Wang et al., 2024) | 2024 | English | ✗ | 32 | 1,211,186 |
| Speech-Forensics (Ji et al., 2024) | 2024 | English | ✗ | - | 7,362 |
| IndicFake (Proposed) | 2025 | 18 Languages | ✓ | 4 | 4,222,759 |

or extracted representations (Kawa et al., 2023; Tak et al., 2021a; Jung et al., 2022). Although effective on standardized benchmarks, English-trained models falter in multilingual scenarios (Korshunov & Marcel, 2016; Müller et al., 2022) or those with accented speech (Ranjan et al., 2024), underscoring the need for cross-lingual methods (Ba et al., 2023). Challenges such as partial-truth detection, explainability, and noise robustness (Ranjan et al., 2023) highlight the urgency for comprehensive datasets and unified architectures for reliable real-world deepfake detection.

**LLM Reprogramming for Non-linguistic Tasks.** Recent work has demonstrated that large pre-trained language models can be reprogrammed to handle non-linguistic modalities by adapting their input or intermediate representations, rather than retraining from scratch. Jin et al. (2024) proposed Time-LLM, which reprograms a frozen LLM for time-series forecasting by transforming numerical inputs into token sequences. Melnyk et al. (2023) introduced ReprogBERT, applying a mapping between biological sequences and BERT embeddings for antibody modeling. Similarly, Fan et al. (2024) presented LLMAir, which adapts LLMs for air-quality prediction through task-specific reprogramming. These studies emphasize the LLM's potential as a universal sequence modeling engine rather than a purely linguistic system—a perspective adopted in SAFARI-LLM, where the LLaMA backbone serves as a multimodal fusion layer integrating semantic and acoustic embeddings for deepfake detection.

## 1.2 Problem Formulation and Research Contributions

The multilingual audio deepfake detection problem involves classifying an audio sample $X \in \mathbb{R}^T$ (where $T$ is the temporal dimension) as real ($y = 0$) or fake ($y = 1$) across languages $l \in L = \{l_1, l_2, \ldots, l_M\}$ and generation methods $m \in M$. Linguistic diversity introduces complexity, as languages exhibit distinct phonetic, acoustic, and prosodic characteristics. Additionally, varied synthesis methods produce unique artifacts, complicating detection in real-world scenarios.

The objective is to minimize the binary cross-entropy loss:

$$\min_{\theta} \mathcal{L}(\theta) = -\frac{1}{N} \sum_{i=1}^{N} [y_i \log(f_\theta(X_i)) + (1 - y_i) \log(1 - f_\theta(X_i))] \tag{1}$$

where $f_\theta$ is the detection model parameterized by $\theta$, and $N$ is the number of training samples. However, the assumption that training and testing data share the same distribution often fails in multilingual settings, where models trained on one language must generalize to others with differing acoustic and linguistic properties. This paper addresses three key research questions to tackle these challenges:

**RQ1 (Cross-Lingual Generalization):** Can a model trained on language $l_a$ detect deepfakes in language $l_b$ ($f_\theta : X^{(l_a)} \rightarrow y^{(l_b)}$)?

This question evaluates whether models can generalize across languages with distinct phonetic inventories, prosodic patterns, and acoustic traits. Success in this task requires learning language-agnostic features, enabling practical deployment where labeled data for every language is unavailable.

**RQ2 (Cross-Language Family Generalization):** Can a model trained on language family $F_a$ detect deepfakes in family $F_b$ ($f_\theta : X^{(F_a)} \to y^{(F_b)}$)?
This extends RQ1 to more diverse linguistic structures, such as Indo-European versus Dravidian families, which differ in phonological systems, word order, and consonant-vowel patterns. This is critical for regions like India, where multiple language families coexist. Success indicates the model captures universal deepfake artifacts across fundamentally different linguistic domains.

**RQ3 (Impact of Model Architecture):** How do architectural choices and input representations $R \in \{R_{\mathrm{raw}}, R_{\mathrm{spec}}, R_{\mathrm{dual}}\}$ affect detection performance ($f_{\theta,R} : X \to y$)?
This investigates the role of raw waveforms ($R_{\mathrm{raw}}$), spectral features ($R_{\mathrm{spec}}$), and dual-stream representations ($R_{\mathrm{dual}}$) in multilingual detection. It also explores how transformer-based encoders, attention mechanisms, and feature fusion impact robustness. Optimal performance requires co-designing architectures with input representations to capture both temporal and frequency-domain cues.

We address these research questions with two primary contributions, significantly advancing multilingual audio deepfake detection:

- **IndicFake Dataset:** We introduce a multilingual audio deepfake dataset with over 4 million samples across 18 languages from Indo-European, Dravidian, and Sino-Tibetan families. Unlike existing datasets that primarily focus on English or have limited multilingual settings, IndicFake enables robust evaluation of cross-lingual and cross-family generalization (RQ1, RQ2). It includes authentic and synthetic audio from state-of-the-art text-to-speech systems, reflecting diverse speakers, acoustic conditions, and generation methods.

- **SAFARI-LLM Architecture:** We propose a novel detection framework combining dual-stream semantic (Whisper) and acoustic (m-HuBERT) encoders via an Audio Feature Unification Module (AFUM). Designed for RQ3, SAFARI-LLM uses LoRA-based fine-tuning and dynamic routing to adaptively integrate semantic and acoustic features, enhancing generalization across languages and generation methods.

- SAFARI-LLM achieves 94.21% accuracy in English-to-Japanese transfer on WaveFake, 84.48% in English-to-Chinese transfer on DECRO, and balanced performance across IndicFake's diverse language families, demonstrating its effectiveness in addressing real-world multilingual deepfake detection challenges.

## 2 Proposed *IndicFake* Dataset

Deepfake detection systems predominantly trained on English audio data struggle when applied to non-English scenarios, emphasizing a critical gap in existing multilingual datasets (Wang et al., 2020; Yamagishi et al., 2021; Yi et al., 2022; 2023). To bridge this gap, we introduce **IndicFake**, an extensive multilingual dataset explicitly designed to enhance robust cross-lingual deepfake detection. IndicFake comprises over **4 million** audio samples, covering English and **17** Indian languages from three major language families: Indo-Aryan, Dravidian, and Sino-Tibetan, as detailed in Table 2. The dataset uniquely integrates authentic speech recordings alongside synthetic audio generated by advanced text-to-speech (TTS) models, offering comprehensive coverage of linguistic nuances and acoustic variations.

### 2.1 Dataset Construction

Creating a multilingual speech dataset for Indian languages is challenging due to the extensive diversity in scripts, phonetic inventories, and prosodic structures. For instance, even languages within the same family, such as Hindi and Marathi, share the same script (Devanagari), whereas languages like Bengali and Punjabi utilize entirely different writing systems (Eastern Nagari and Gurmukhi, respectively). Similarly,

Table 2: Characteristics of the proposed IndicFake dataset. The dataset contains real and synthetic speech samples across English and 17 Indian languages, including per-model TTS splits by gender and overall dataset composition.

| Language | Real Data | MMS Male | IndicTTS Male | IndicTTS Female | DonaLabTTS Male | DonaLabTTS Female | DonaLabTTS2 Male | DonaLabTTS2 Female |
|---|---|---|---|---|---|---|---|---|
| Assamese | 112,426 | 30,000 | 29,982 | 29,982 | – | – | 26,927 | 26,927 |
| Bangla | 111,077 | 30,000 | 29,986 | 29,986 | 29,640 | 29,640 | 29,653 | 29,653 |
| Bodo | 5,715 | – | – | – | – | – | – | 22,118 |
| Dogri | 3,649 | 5,499 | – | – | – | – | – | – |
| English | | 30,000 | 29,908 | 29,908 | – | – | 30,043 | 21,631 |
| Gujarati | 144,337 | – | – | – | 28,885 | 28,885 | 29,261 | 29,261 |
| Hindi | 221,022 | 30,000 | 29,915 | 29,915 | 29,186 | 27,345 | 29,508 | 15,188 |
| Kannada | 214,855 | 30,000 | 29,995 | 29,995 | 22,476 | 28,793 | 29,240 | 29,240 |
| Maithili | 328 | 14,960 | – | – | – | – | – | – |
| Malayalam | 153,954 | 30,000 | 29,994 | 29,994 | 28,851 | 28,851 | 28,738 | 8,778 |
| Manipuri | 46,813 | – | 4,816 | 4,815 | – | – | 18,553 | – |
| Marathi | 211,906 | 30,000 | 26,306 | – | 29,376 | 29,376 | 29,429 | 29,429 |
| Oriya | 115,732 | 30,000 | 27,319 | 27,318 | – | – | 29,724 | 29,722 |
| Punjabi | 137,442 | 30,000 | 24,751 | 24,604 | – | – | 30,033 | 30,033 |
| Rajasthani | – | – | 4,925 | 4,926 | 4,926 | 4,926 | 4,926 | 4,925 |
| Tamil | 146,215 | 30,000 | 29,920 | 29,920 | 25,134 | 28,284 | 30,002 | 30,002 |
| Telugu | 259,908 | 30,000 | 29,989 | – | 28,810 | 28,810 | 29,558 | 29,536 |
| Urdu | 112,185 | 41,335 | – | – | – | – | 30,000 | 30,000 |
| **#samples** | 1,997,564 | 391,794 | 327,806 | 271,363 | 227,284 | 234,910 | 405,595 | 366,443 |
| **#samples/model** | 1,997,564 | 391,794 | 599,169 | | 462,194 | | 772,038 | |
| **#samples/class** | 1,997,564 | 2,225,195 | | | | | | |
| **Total** | **4,222,759** | | | | | | | |

the Dravidian languages (Kannada, Malayalam, Tamil, Telugu) each possess distinctive scripts, and Sino-Tibetan languages (Manipuri, Bodo) frequently adopt multiple scripts (e.g., Eastern Nagari, Devanagari). Recognizing this complexity, our dataset construction approach involves two distinct steps: the systematic collection of authentic speech from diverse linguistic and script backgrounds, and the generation of synthetic speech samples that capture a wide range of tonal and phonetic characteristics.

**Real Data Collection:** IndicFake derives its authentic speech exclusively from the curated Dhwani corpus (Javed et al., 2022), which provides lists of Creative Commons-licensed YouTube URLs specifically prepared for ASR research. From Dhwani, we selected 17 languages that cover the Indo-European (Assamese, Bangla, Marathi, Oriya, Punjabi, Rajasthani, Maithili, Dogri, Urdu, Gujarati, Hindi), Dravidian (Kannada, Malayalam, Tamil, Telugu), and Sino-Tibetan (Manipuri, Bodo) language families. We sampled 200 CC-licensed videos per language (where available) across the education, news, technology, sports, and finance domains to capture a variety of speaking styles.

All recordings were converted to 16 kHz mono-channel format. We applied Voice Activity Detection (py-webrtcvad, aggressiveness=2) to remove non-speech segments. We filtered out segments with Signal-to-Noise Ratios below 15 dB using the WADA-SNR method, ensuring consistent audio quality. Each recording was segmented into non-overlapping ~5-second windows; final clip lengths vary as VAD trims boundaries, and very short segments are discarded. To minimize the risk of synthetic contamination in the real subset, each source video underwent independent review by two annotators, following standardized guidelines. Annotators excluded: (a) content explicitly labeled as "TTS," "AI voice," or "synthetic" in titles/descriptions/channels; (b) dubbed or post-processed material; (c) content exhibiting synthetic speech markers such as unnaturally

monotonic prosody, robotic cadence, or suspiciously uniform noise floors. We constructed a 70/30 train-test split at the *video* level to ensure speaker independence and prevent content leakage across partitions, yielding approximately 2,660 hours of verified authentic speech for training and evaluation.

**Fake Audio Generation:** Synthetic speech samples within IndicFake were generated using multiple sophisticated TTS models. These models span a variety of architectures, from traditional pipeline-based approaches to state-of-the-art end-to-end frameworks, ensuring comprehensive coverage of synthetic speech characteristics:

- **IndicTTS (F01)** (Kumar et al., 2022): Utilizes FastPitch (Ren et al., 2019) for efficient mel-spectrogram prediction coupled with HiFi-GAN (Kong et al., 2020) for generating high-fidelity audio waveforms. This combination ensures rapid generation of natural-sounding speech with accurate prosodic modeling.

- **DonaLabTTS (F02) & DonaLabTTS2 (F03)** (Ren et al., 2020): Both models are derived from the FastSpeech2 framework but differ significantly in their phoneme alignment strategies. DonaLabTTS applies a hybrid segmentation method to achieve robust phoneme-level alignments. At the same time, DonaLabTTS2 utilizes the precise Montreal Forced Aligner (MFA) method, resulting in consistently accurate duration modeling across diverse linguistic contexts.

- **Massive Multilingual Speech (F04)** (Pratap et al., 2024): Employs a Variational Inference with adversarial learning for Text-to-Speech (VITS) model, directly generating raw waveforms without intermediate spectrogram stages. This approach effectively captures an expansive range of prosodic variations, benefiting from extensive multilingual pre-training encompassing up to 1,100 languages.

Table 3: Language metadata across the IndicFake dataset showing language codes, speaker gender distribution, script systems, language families, and native regions, highlighting the dataset's linguistic and demographic diversity.

| Language | Code | Speakers | Script | Family | Native Region |
|---|---|---|---|---|---|
| Assamese | as | male, female | Eastern-Nagari | Indo-European | Assam |
| Bangla | bn | male, female | Eastern-Nagari | Indo-European | West-Bengal, Bangladesh |
| Bodo | brx | female | Devanagari | Sino-Tibetan | Bodoland Territory |
| Dogri | dgo | male | Dogri | Indo-European | Rajasthan |
| English | en | male, female | English | Indo-European | Pan India |
| Gujarati | gu | male, female | Gujarati | Indo-European | Gujarat |
| Hindi | hi | male, female | Devanagari | Indo-European | Hindi Belt |
| Kannada | kn | male, female | Kannada | Dravidian | Karnataka |
| Maithili | ma | male | Devanagari | Indo-European | Bihar |
| Malayalam | ml | male, female | Malayalam | Dravidian | Kerala |
| Manipuri | mni | male, female | Meetei, Eastern-Nagari | Sino-Tibetan | Imphal valley (Manipur) |
| Marathi | mr | male, female | Devanagari | Indo-European | Maharashtra |
| Oriya | or | male, female | Odia | Indo-European | Odisha |
| Punjabi | pa | male, female | Gurumukhi | Indo-European | Eastern-Punjab |
| Rajasthani | raj | male, female | Devanagari | Indo-European | Rajasthan |
| Tamil | ta | male, female | Tamil | Dravidian | Tamil Nadu |
| Telugu | te | male, female | Telugu | Dravidian | Andhra Pradesh, Telangana |
| Urdu | ur | male, female | Arabic | Indo-European | Hindi Belt |

## 2.2 Dataset Diversity

The IndicFake dataset is a comprehensive and linguistically diverse collection featuring 18 languages across India's three major language families. This extensive diversity ensures robust cultural and linguistic rep-

Figure 2: Distribution of audio samples across 18 languages, showing representation of major languages and inclusion of low-resource languages.

resentation, essential for effective multilingual deepfake detection. The dataset includes various scripts, reflecting India's rich textual heritage. For instance, the Eastern-Nagari script is employed for Assamese and Bangla, representing linguistic traditions from eastern India. The widely used Devanagari script encompasses Hindi, Marathi, and Rajasthani, illustrating central and western linguistic characteristics. Southern languages—Tamil, Telugu, Malayalam, and Kannada—each possess distinctive scripts with unique characters and writing conventions.

Gender representation within the dataset has been carefully curated to maintain balanced voice diversity across most languages. Both male and female voices are comprehensively represented, supporting nuanced analyses of gender-specific vocal features. Languages like Dogri, Maithili, and Bodo exhibit single-gender representation due to demographic constraints and data availability limitations in these linguistic communities.

Categorizing languages by family provides essential linguistic context. The Indo-European family comprises twelve languages, including Assamese, Bangla, Dogri, English, Gujarati, Hindi, Maithili, Marathi, Oriya, Punjabi, Rajasthani, and Urdu, highlighting the significant diversity within this linguistic group. The Dravidian family, represented by languages such as Kannada, Malayalam, Tamil, and Telugu, showcases the distinct linguistic identity of southern India. The inclusion of Sino-Tibetan languages, such as Bodo and Manipuri, adds further linguistic depth to the dataset.

Geographically, IndicFake captures linguistic diversity across India, ranging from the northern mountainous regions (Dogri) to the tropical southern landscapes (Malayalam), and from the western states (Gujarati) to the northeastern areas (Assamese). This comprehensive geographic coverage ensures broad cultural representation, effectively reflecting the linguistic richness and complexity of the Indian subcontinent. Detailed dataset information is presented in Table 3.

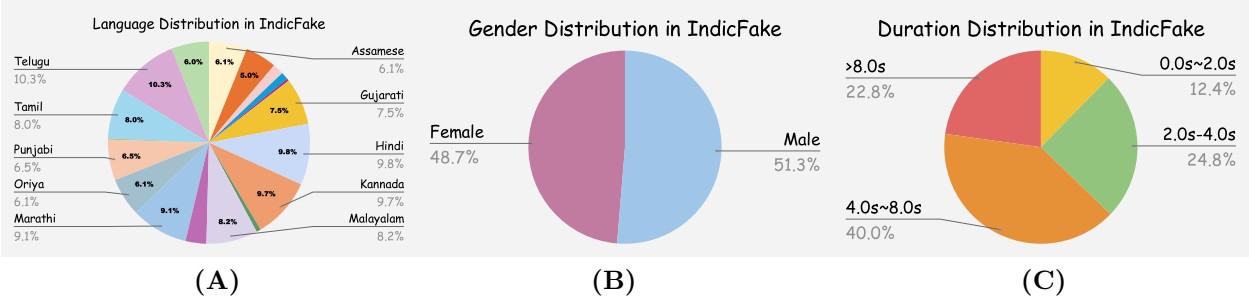

Figure 3: Showcases the distribution of speaker gender, language, and duration in the IndicFake dataset, demonstrating dataset balance. The language distribution (A) shows balanced coverage across 18 languages. The speaker distribution (B) highlights maintained gender balance across both real and synthetic speech samples, enhancing the dataset's representativeness for deepfake detection research. While the duration distribution (C) indicates natural variation, ranging from 0.5 to over 8 seconds, it reflects real-world speech patterns.

## 2.3 Dataset Statistics

The IndicFake dataset comprises over 4.2 million speech samples, totaling approximately 7,350 hours of audio data. This extensive collection spans English and 17 Indian languages, grouped into three prominent language families: Indo-European, Dravidian, and Sino-Tibetan. A detailed breakdown of this dataset is presented in Table 2. IndicFake maintains a balanced distribution, with most languages exceeding 100,000 samples, while thoughtfully preserving representation for low-resource languages. Specifically, the dataset comprises approximately 2.2 million synthetic audio samples (approximately 4,690 hours) and 2 million real audio samples (approximately 2,660 hours), both generated using four advanced TTS systems.

The language distribution within IndicFake demonstrates deliberate resource allocation to ensure robust representativeness. Major languages such as Hindi (414,594 samples), Kannada (412,079 samples), and Telugu (436,611 samples) are well-represented, aligning with their widespread usage and significant speaker populations. Medium-resource languages, including Malayalam (349,477 samples), Tamil (339,160 samples), and Marathi (385,822 samples), also maintain strong representation, ensuring comprehensive analytical capabilities. Crucially, IndicFake incorporates lower-resource languages, such as Bodo (74,997 samples), Dogri (50,545 samples), and Rajasthani (9,148 samples), highlighting the dataset's inclusive design, which aims to support technology solutions across diverse language communities, regardless of their size or resource availability. Figure 2 provides a visual overview of the dataset's language-wise distribution.

The duration of audio samples within IndicFake has been carefully curated to encompass a range of speech scenarios. Short audio segments (0.8–2.0 seconds) constitute 12.4% of the dataset, effectively capturing brief utterances and quick speech interactions. Medium-length segments (2.0–4.0 seconds) represent typical conversational turns, accounting for 24.8% of the dataset. Longer segments (4.0–8.0 seconds), comprising 40.0%, offer substantial context suitable for detailed analysis. Finally, extended segments exceeding 8.0 seconds make up 22.8%, enabling exploration of longer speech patterns, prosody, and extended conversational contexts.

IndicFake also achieves near-perfect gender parity, with male speakers representing 51.3% and female speakers comprising 48.7% of the total audio samples. This balanced gender representation is essential for developing unbiased audio processing and deepfake detection algorithms that can achieve robust performance across diverse speaker demographics. Figure 3 illustrates the distribution across languages, durations, and gender, highlighting the dataset's comprehensive and balanced nature.

## 2.4 Dataset Comparison

To contextualize IndicFake's contribution within the existing landscape of audio deepfake datasets, we conducted a comparative analysis using Jaccard similarity indices and UpSet plot visualizations. Jaccard simi-

Table 4: Jaccard similarity indices comparing language overlap between IndicFake and existing audio deep-fake datasets, demonstrating IndicFake's unique contribution to language coverage in deepfake detection research.

| Dataset | ASVspoof 2015 | ASVspoof 2019-LA | FoR | ASVspoof 2021-LA | ASVspoof 2021-DF | WaveFake | ADD2022-LF | Latin American | CFAD | DECRO | ASVspoof5 | Speech-Forensics | MLAAD |
|---|---|---|---|---|---|---|---|---|---|---|---|---|---|
| **Jaccard Index** | 0.06 | 0.06 | 0.06 | 0.06 | 0.06 | 0.05 | 0.00 | 0.00 | 0.00 | 0.05 | 0.06 | 0.06 | 0.06 |

larity indices revealed minimal overlap between IndicFake and existing datasets, ranging from 0.00 to 0.06. IndicFake shares the highest overlap (0.06) with datasets such as ASVspoof 2015, ASVspoof 2019-LA, FoR, ASVspoof 2021-LA, ASVspoof 2021-DF, and Speech-Forensics. This notably low overlap underscores IndicFake's distinctiveness, particularly in terms of language diversity. A detailed comparison using Jaccard indices is provided in Table 4.

The UpSet plot visualization in Figure 4 offers further insights into the dataset intersections. MLAAD emerges as the most linguistically diverse dataset with 38 languages, closely followed by IndicFake's substantial coverage of 18 languages. The most significant intersection occurs between MLAAD and IndicFake, highlighting overlapping coverage of several Indian languages. However, this intersection remains comparatively small relative to each dataset's total linguistic scope, reinforcing the complementary nature of these resources. Other datasets, such as DECRO and WaveFake, each intersect minimally, emphasizing their narrower linguistic coverage. Most other existing datasets primarily concentrate on English or Chinese, with minimal overlap across languages.

This comprehensive analysis highlights IndicFake's unique and significant contribution to linguistic diversity within the field of audio deepfake research. By encompassing numerous underrepresented Indian languages, IndicFake fills a critical gap in existing datasets, establishing itself as a valuable resource for developing more inclusive, robust, and universally applicable deepfake detection technologies.

Table 5: Comparing speech quality metrics for real and fake audio samples of the proposed IndicFake dataset. SIG: Speech Quality, BAK: Background Noise Quality, OVRL: Overall Quality, P808-MOS: ITU-T P.808 Mean Opinion Score

| Subset | SIG | BAK | OVRL | P808-MOS |
|---|---|---|---|---|
| Real | 3.175 | 3.367 | 2.650 | 3.233 |
| Fake | 3.440 | 4.111 | 3.190 | 3.879 |

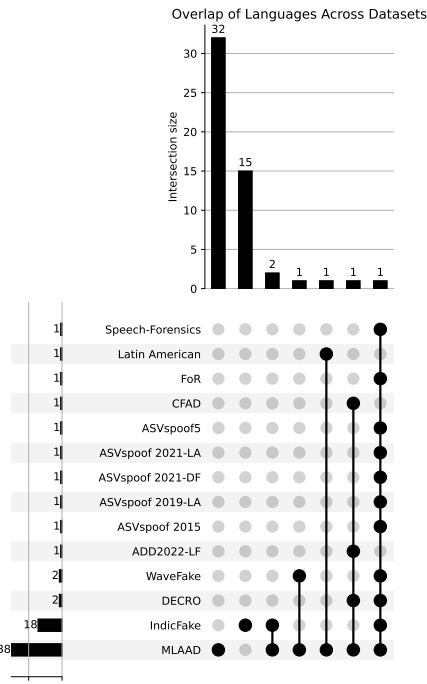

Figure 4: UpSet plot visualizing the intersection of languages across audio deepfake datasets. The plot reveals limited overlap between datasets, with MLAAD (38 languages) and IndicFake (18 languages) showing the highest language diversity.

## 2.5 Dataset Quality

The quality evaluation of the IndicFake dataset provides essential insights into the perceptual characteristics of real and synthetic audio samples. We employ four key metrics: Speech Quality (SIG), Background Noise Quality (BAK), Overall Quality (OVRL) (Reddy et al., 2022), and the ITU-T P.808 Mean Opinion Score (MOS) (Reddy et al., 2021). Table 5 summarizes these results. Synthetic audio samples demonstrate superior performance in background noise quality (BAK: 4.111 synthetic vs. 3.367 real) and overall quality (OVRL: 3.19 synthetic vs. 2.650 real), indicating effective noise suppression. This aligns with prior DNSMOS findings (Reddy et al., 2021; 2022), confirming that noise reduction significantly enhances perceived audio quality.

The 8.3% improvement in speech clarity for synthetic samples (SIG: 3.440 vs. 3.175 for real) suggests synthetic audio effectively maintains phonetic clarity. However, subtle artifacts remain detectable, particularly during specialized analyses. The higher MOS scores (3.879 synthetic vs. 3.233 real) further confirm synthetic audio's human-like perceptual quality, mirroring observations in multilingual deepfake detection research.

These results present a dual challenge for detection systems. Synthetic audio achieves a quality sufficient to deceive casual listeners, as evidenced by elevated MOS and OVRL scores; yet, it retains identifiable artifacts detectable through structured analysis. Notably, the BAK metric underscores significant improvements in noise suppression (21.3% increase). In contrast, the narrower margin in SIG (8.3% improvement) highlights advancements in phonetic fidelity but points toward lingering subtle synthetic artifacts. This quality paradox highlights the need for detection methods that focus on residual artifacts, rather than relying solely on conventional quality indicators. IndicFake's detailed quality evaluation thus offers a comprehensive framework to drive the development of robust deepfake detection systems.

## 2.6 Dataset Protocol

To facilitate rigorous and systematic evaluations, IndicFake is structured into three distinct subsets. Set A encompasses ten Indo-European languages: Assamese, Bengali, Dogri, Gujarati, Hindi, Maithili, Marathi, Odia, Punjabi, and Urdu. Set B includes four Dravidian languages: Kannada, Malayalam, Tamil, and Telugu. Set C comprises Bodo, Manipuri, and English.

For Sets A and B, we implement train-test splits to ensure speaker and model independence. Training datasets contain synthetic samples from DonaLabTTS2 and MMS TTS, while evaluation datasets include synthetic samples from DonaLabTTS and IndicTTS, facilitating evaluation of unseen TTS models. Additionally, real speech data is partitioned to maintain speaker independence and avoid biases. Set C is exclusively designated for cross-lingual generalization testing, featuring languages entirely unseen during training. This structured protocol supports comprehensive evaluation across three dimensions: cross-model, cross-language, and speaker generalization, thereby establishing robust benchmarks for multilingual deepfake detection systems.

## 2.7 Dataset Spectral Analysis

To understand the spectral characteristics of synthetic speech in IndicFake, we conducted a detailed frequency analysis. Figure 5 illustrates average energy distributions across frequency bands, alongside difference plots highlighting deviations from natural speech. The spectral profiles of real audio reveal typical characteristics, with prominent energy concentrated in lower frequencies (0–3 kHz) and gradual declines at higher frequencies. Synthetic audio generated by MMS, IndicTTS, DonaLabTTS, and DonaLabTTS2 maintain similar overall spectral shapes, but exhibit notable deviations, particularly within higher frequency bands (6–11 kHz).

Difference plots quantify these spectral deviations explicitly. MMS audio exhibits the most significant high-frequency artifacts, showing variations of up to $\pm10$ dB compared to natural speech. DonaLabTTS2 achieves improved spectral fidelity over its predecessor, especially in mid-range frequencies (3–6 kHz), though some discrepancies persist at higher frequencies. IndicTTS maintains more consistent spectral behavior but still exhibits notable deviations above 6 kHz.

These characteristic spectral differences between synthetic and natural speech provide reliable indicators for deepfake detection systems. Persistent high-frequency artifacts across all TTS systems suggest fundamental

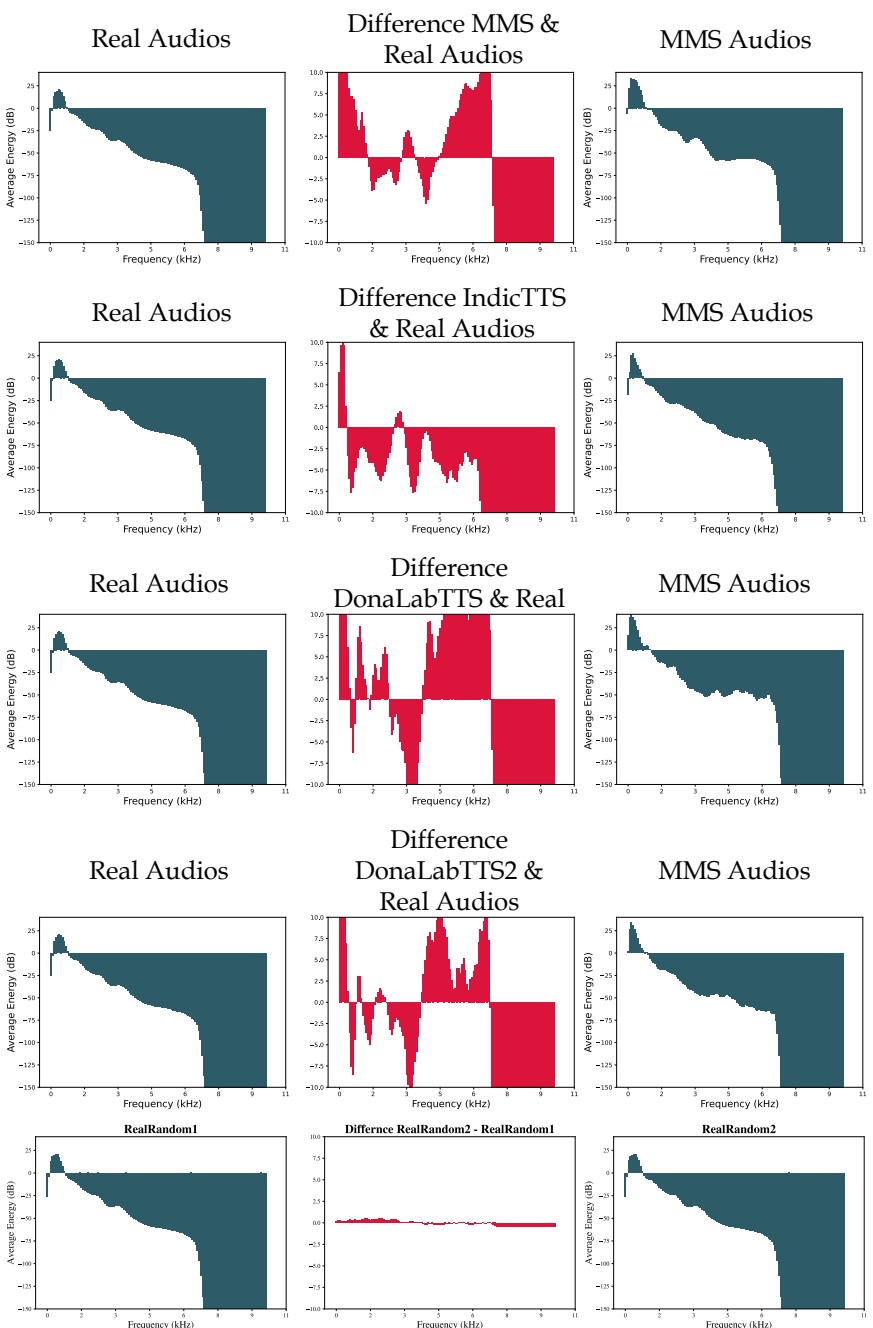

Figure 5: Spectral comparison between real and synthetic speech across different TTS systems. Each row shows the average energy distribution (dB) across frequency bins for real speech (left), synthetic speech (right), and their difference (center), highlighting characteristic deviations in the high-frequency region (6-11 kHz). The last plot shows the difference between two random real sets. These plots are inspired by the Wavefake paper(Frank & Schönherr, 2021).

limitations in current synthetic speech generation methods. These insights highlight both opportunities to enhance synthetic speech quality and strategies to improve deepfake detection techniques.

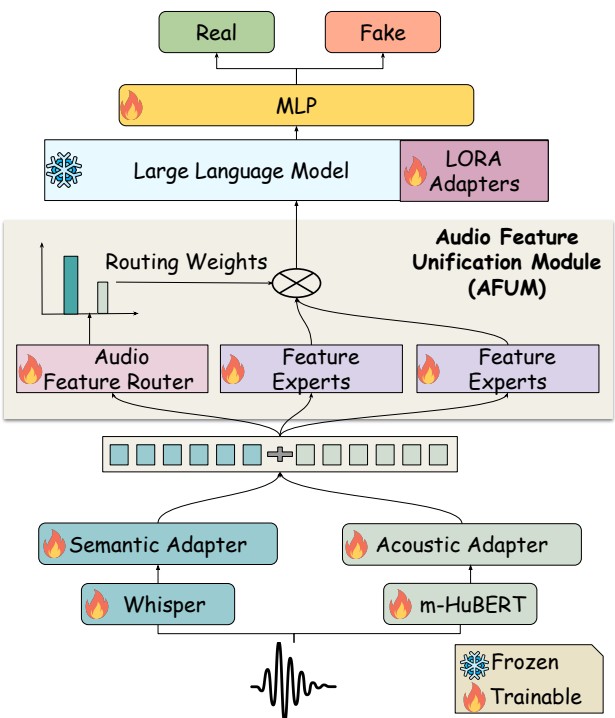

Figure 6: Overview of the proposed approach. Two speech encoders and adapters with different focuses are utilized, where Whisper and its corresponding adapter are used for extracting semantic information, and m-HuBERT is used for extracting acoustic information. Before being fed to the LLM, these two representations are concatenated together.

## 3 Proposed SAFARI-LLM

The proposed **SAFARI-LLM** (**S**emantic **A**coustic **F**eature **A**daptive **R**outer with **I**ntegrated **LLM**) addresses three key research questions: cross-lingual generalization (RQ1), extended cross-language family generalization (RQ2), and the impact of model architecture on performance (RQ3). SAFARI-LLM integrates semantic and acoustic speech processing with a Large Language Model (LLM) to enable robust, multilingual deepfake detection across diverse linguistic contexts. As depicted in Figure 6, SAFARI-LLM employs a dual-stream architecture comprising two specialized encoders: Whisper (Radford et al., 2022) for semantic analysis and m-HuBERT (Boito et al., 2024) for acoustic profiling. Their outputs are fused using an Audio Feature Unification Module (AFUM), which dynamically balances semantic and acoustic features. The unified representation is then processed by an LLM, fine-tuned with Low-Rank Adaptation (LoRA), to achieve high detection accuracy across varied linguistic settings.

### 3.1 Dual-Stream Speech Encoders

The dual-stream architecture addresses RQ1 and RQ2 by capturing both semantic and acoustic information critical for effective cross-lingual and cross-language family deepfake detection. Whisper-large (Radford et al., 2022), pretrained on 96 languages, extracts high-level semantic content from audio inputs, enabling robust generalization across languages. Concurrently, m-HuBERT-base (Boito et al., 2024), pretrained on 147 languages, captures fine-grained acoustic features, including speaker identity, timbre, and prosodic patterns, which are essential for detecting subtle deepfake artifacts.

Formally, given a batch of audio signals $\mathbf{X} \in \mathbb{R}^{B \times T}$, where $B$ is the batch size and $T$ is the temporal dimension, we first transform each signal into a log-mel spectrogram $\mathbf{S} \in \mathbb{R}^{B \times F \times T}$, where $F$ denotes the

frequency dimension. Semantic embeddings are computed as:

$$\mathbf{H}_s = \text{Whisper}(\mathbf{S}), \quad \mathbf{H}_s \in \mathbb{R}^{B \times T_s \times D_s}, \tag{2}$$

Where $T_s$ is the temporal dimension of the semantic features, and $D_s$ is the embedding dimension. Acoustic embeddings are derived using m-HuBERT:

$$\mathbf{H}_a = \text{mHuBERT}(\mathbf{X}), \quad \mathbf{H}_a \in \mathbb{R}^{B \times T_a \times D_a}, \tag{3}$$

Where $T_a$ and $D_a$ represent the temporal and embedding dimensions of the acoustic features, respectively. To unify these heterogeneous embeddings, we employ adapter modules that perform the following operations:

1. Apply two 1D convolutional layers to reduce dimensionality and align temporal resolutions between $\mathbf{H}_s$ and $\mathbf{H}_a$.

2. Utilize a bottleneck adapter (Houlsby et al., 2019) to balance computational efficiency and feature expressiveness.

3. Project both embeddings into a shared dimensional space using a linear layer.

The adapted embeddings, $\mathbf{H}'_s$ and $\mathbf{H}'_a$, are mapped to a common space:

$$\mathbf{H}'_s, \mathbf{H}'_a \in \mathbb{R}^{B \times 38 \times 1024}. \tag{4}$$

These embeddings are concatenated to form a unified input:

$$\mathbf{x}_m = [\mathbf{H}'_s, \mathbf{H}'_a], \quad \mathbf{x}_m \in \mathbb{R}^{B \times 38 \times 2048}. \tag{5}$$

### 3.2 Audio Feature Unification Module

The AFUM addresses RQ3 by dynamically balancing the contributions of semantic and acoustic features to optimize detection performance. AFUM comprises $K$ projection experts $\{P_k\}$, each implemented as a transformer-based layer, and a multi-layer perceptron (MLP) Audio Feature Router $R$ (Puigcerver et al., 2023). This design enables adaptive feature weighting, ensuring that the model prioritizes relevant information based on the input audio characteristics.

Given the concatenated input $\mathbf{x}_m \in \mathbb{R}^{B \times L \times D}$, where $L = 38^4$ and $D = 2048$, AFUM computes a unified representation as a weighted sum of expert outputs:

$$\bar{\mathbf{x}}_m = \sum_{k=1}^{K} w_{m,k} \cdot P_k(\mathbf{x}_m), \tag{6}$$

where $w_{m,k}$ are the routing weights for the $k$-th expert, and $P_k(\mathbf{x}_m)$ denotes the output of the $k$-th projection expert. The routing weights are computed dynamically by the router $R$:

$$\mathbf{w}_m = \sigma(R(\mathbf{x}_m)), \quad \mathbf{w}_m \in \mathbb{R}^{B \times L \times K}, \tag{7}$$

Where $\sigma(\cdot)$ is the softmax function, ensuring that the weights are normalized across the $K$ experts for each input token. This mechanism enables AFUM to adaptively emphasize either semantic or acoustic features based on the input, thereby enhancing robustness across diverse linguistic contexts.

---

[4]The temporal dimension of 38 is directly adopted from the Whisper model's semantic embeddings, and the acoustic embeddings are aligned to this dimension.

### 3.3 Large Language Model (LLM) Integration and Classification

To integrate semantic and acoustic embeddings with multilingual priors and long-range attention, we use a LoRA-adapted LLaMA-7B backbone as a *sequence integrator over audio embeddings*. The LLM operates on the unified sequence $\bar{\mathbf{x}}_m$ output by AFUM—formed from Whisper (semantic) and m-HuBERT (acoustic) embeddings after their respective adapters—*not* on text tokens or transcripts. This design enables the backbone to model token-level cross-modal dependencies and alignments between semantic content and its acoustic realization.

**Instantiation and adaptation.** We instantiate the backbone with LLaMA–7B (Touvron et al., 2023) enhanced via Vicuna instruction-following fine-tuning (Vicuna, 2023), and adapt it efficiently using Low-Rank Adaptation (LoRA) (Hu et al., 2021). LoRA adapters with rank $r=8$ and scaling $\alpha=16$ are inserted into the *query and value* projections of all self-attention layers; the base transformer weights of LLaMA–7B are kept frozen, preserving pretrained priors while enabling task-specific adaptation.

The unified embeddings $\bar{\mathbf{x}}_m$ from AFUM are processed by the LoRA-adapted LLM:

$$\mathbf{y}_{\text{LLM}} = \text{LLM}_{\text{LoRA}}(\bar{\mathbf{x}}_m). \tag{8}$$

The resulting embeddings are fed into a Multi-Layer Perceptron (MLP) for binary classification:

$$\hat{y} = \sigma(\text{MLP}(\mathbf{y}_{\text{LLM}})), \tag{9}$$

where $\sigma(\cdot)$ is the sigmoid activation function, producing a probability score for the binary classification task (real vs. fake audio).

**Why an LLM?** We adopt a LoRA-adapted LLaMA-7B model as a pretrained multilingual sequence integrator whose long-range attention enables effective fusion of semantic (Whisper) and acoustic (m-HuBERT) embeddings. Our motivation for incorporating an LLM is not to exploit its linguistic knowledge or text generation capabilities, but rather to leverage its general sequence modeling capacity to align heterogeneous modalities. This design choice is consistent with the emerging paradigm of LLM reprogramming, which demonstrates that pretrained language models can be repurposed for non-linguistic domains such as time-series forecasting or biological sequence modeling by adapting inputs into the model's latent space ( Jin et al. (2024); Fan et al. (2024); Melnyk et al. (2023)

In this view, the LLM serves as a flexible and high-capacity transformer backbone for multimodal integration, rather than acting as a linguistic expert. The LoRA fine-tuning enables efficient task-specific adaptation while preserving the general cross-domain priors that promote robust alignment between content and acoustics across languages and deepfake synthesis methods.

## 4 Experimental Setup and Protocols

This section outlines the experimental setup, detailing the datasets used, baseline models for comparison, implementation specifics, evaluation metrics, and the structured protocols designed to comprehensively assess our proposed SAFARI-LLM model.

**Existing Datasets** Apart from the proposed IndicFake corpus, we evaluate our method using two prominent multilingual datasets to thoroughly examine cross-lingual and cross-synthesis generalization capabilities:

- **DECRO** (Ba et al., 2023): Contains English and Chinese subsets, with 21,218 bona fide Chinese samples and 12,484 English samples, each with predefined training, development, and evaluation partitions.

- **WaveFake** (Frank & Schönherr, 2021): Features 136,085 samples, including 121,085 in English and 15,000 in Japanese, designed explicitly for assessing multilingual generalization and synthesis variability. We follow the leave-one-out protocol enforcing evaluation on an unseen generator. For

Table 6: Comparison of deepfake detection performance across English and Japanese languages using the WaveFake dataset, showing cross-lingual generalization capabilities for different model architectures. Acc: Accuracy (%), EER: Equal Error Rate.

| Models | Train on English, Eval on English | | Train on English, Eval on Japanese | | Train on Japanese, Eval on English | | Train on Japanese, Eval on Japanese | |
|---|---|---|---|---|---|---|---|---|
| | Acc | EER(%) | Acc | EER(%) | Acc | EER(%) | Acc | EER(%) |
| Whisper MesoNet | 10.82 | 37.62 | 33.33 | 43.80 | 89.18 | 46.40 | 66.67 | 44.89 |
| MesoNet | 89.18 | 0.57 | 66.67 | 3.06 | 89.74 | 15.72 | 79.05 | 5.74 |
| SSLModel | 89.47 | 19.64 | 66.67 | 41.25 | 89.18 | 51.87 | 100.00 | 0.00 |
| Whisper SpecRNet | 89.83 | 24.46 | 67.23 | 33.47 | 50.45 | 36.97 | 89.97 | 6.38 |
| Whisper LCNN | 92.29 | 14.62 | 72.76 | 31.03 | 16.60 | 36.55 | 87.97 | 12.76 |
| Conformer | 93.53 | 8.96 | 54.92 | 45.78 | 89.18 | 46.03 | 99.98 | 0.01 |
| RawNet2 | 99.79 | 0.26 | 66.65 | 48.84 | 18.71 | 42.74 | 99.70 | 0.18 |
| SpecRNet | 99.80 | 0.01 | 84.85 | 3.30 | 52.30 | 6.84 | 99.77 | 0.00 |
| RawGAT-ST | 99.85 | 0.24 | 87.42 | 8.36 | 57.49 | 19.85 | 99.01 | 0.34 |
| AASIST | 99.95 | 0.08 | 89.27 | 6.86 | 12.05 | 27.80 | 91.51 | 0.71 |
| RawBMamba | 99.98 | 0.03 | 83.25 | 2.87 | 37.26 | 14.00 | 99.94 | 0.04 |
| LCNN | 99.98 | 0.02 | 90.92 | 8.27 | 10.82 | 18.01 | 99.95 | 0.06 |
| RawNet3 | 99.99 | 0.03 | 86.67 | 12.10 | 80.53 | 32.06 | 98.93 | 0.91 |
| Whisper-frontend-LCNN | 99.98 | 0.02 | 85.09 | 1.28 | 80.48 | 2.08 | 99.92 | 0.03 |
| **Proposed** | **99.99** | 0.02 | **94.21** | 2.48 | **92.31** | 5.31 | **100.00** | 0.00 |

both English and Japanese, MB-MelGAN is excluded from the training process. Evaluation spans MelGAN, MelGAN(L), FB-MelGAN, MB-MelGAN (unseen), HiFi-GAN, WaveGlow, PWG, and TTS, jointly testing model and linguistic generalization.

**Baseline Models**    We benchmark SAFARI-LLM against a comprehensive set of 15 baseline architectures, categorized based on their input modalities:

- **Raw Waveform Models**: Including RawBMamba Chen et al. (2024), Conformer Rosello et al. (2023), SSLModel Tak et al. (2022), AASIST weon Jung et al. (2022a), RawGAT-ST Tak et al. (2021c), RawNet2 Tak et al. (2021b), and RawNet3 weon Jung et al. (2022b), which operate directly on time-domain signals.

- **Spectrogram-based Models**: LCNN (Wu et al., 2018), MesoNet (Afchar et al., 2018) (specifically the MesoInception-4 variant), and SpecRNet (Kawa et al., 2022a; 2023), which process frequency-domain spectrograms.

For spectrogram-based baselines, we test standard cepstral features (LFCC and MFCC), as well as advanced embeddings from the Whisper encoder, alone and in combination with cepstral features, inspired by insights from Kawa et al. (2022b).

**Evaluation Protocols**    Our evaluation protocols are explicitly structured around the three primary RQs:

**RQ1: Cross-Lingual Generalization.** We train models on one language and test on another within the same dataset, utilizing WaveFake (English-Japanese) and DECRO (English-Chinese). These experiments specifically measure each model's capacity to detect deepfake audio across distinct linguistic domains.

**RQ2: Extended Cross-Language Family Generalization.** To examine generalization across fundamentally different language families, we use subsets from IndicFake: Set A (Indo-European) and Set B (Dravidian). We conduct bi-directional experiments, training on one family and testing on the other. We maintain speaker and synthesis-model independence by employing different TTS models—DonaLabTTS2 and MMS TTS for training, and DonaLabTTS and IndicTTS for evaluation.

Table 7: Equal Error Rate (EER) of the SAFARI-LLM on different subsets (LFCC). We train a new model for each data set and compute the EER.

| Training Set | LJSpeech | | | | | | | | JSUT | |
|---|---|---|---|---|---|---|---|---|---|---|
| | MelGAN | MelGAN (L) | MB-MelGAN | FB-MelGAN | HiFi-GAN | PWG | WaveGlow | TTS | MB-MelGAN | PWG |
| MelGAN | 0 | 0.001 | 0.119 | 0.333 | 0.343 | 0.175 | 0.12 | 0.005 | 0.232 | 0.021 |
| MelGAN (L) | 0 | 0 | 0.313 | 0.551 | 0.512 | 0.398 | 0.161 | 0 | 0.159 | 0.012 |
| MB-MelGAN | 0.007 | 0.014 | 0.001 | 0.027 | 0.129 | 0.027 | 0.06 | 0.05 | 0.075 | 0.03 |
| FB-MelGAN | 0.003 | 0.004 | 0.002 | 0.003 | 0.04 | 0.004 | 0.02 | 0.011 | 0.043 | 0.012 |
| HiFi-GAN | 0.127 | 0.145 | 0.258 | 0.333 | 0.008 | 0.185 | 0.145 | 0.017 | 0.299 | 0.094 |
| PWG | 0.507 | 0.555 | 0.495 | 0.704 | 0.683 | 0.011 | 0.468 | 0.537 | 0.225 | 0.094 |
| WaveGlow | 0.044 | 0.134 | 0.147 | 0.424 | 0.378 | 0.255 | 0 | 0.055 | 0.432 | 0.34 |

Table 8: Comparison of the proposed model with different baseline models under the single training set protocol.

| Training Set | WaveFake GMM (Table 2 ) | WaveFake RawNet2 (Table 3 ) | SAFARI-LLM |
|---|---|---|---|
| MelGAN | 0.215 | 0.292 | 0.135 |
| MelGAN (L) | 0.222 | 0.258 | 0.211 |
| MB-MelGAN | 0.108 | 0.357 | 0.042 |
| FB-MelGAN | 0.062 | 0.363 | 0.014 |
| HiFi-GAN | 0.089 | 0.319 | 0.161 |
| PWG | 0.124 | 0.358 | 0.428 |
| WaveGlow | 0.085 | 0.294 | 0.221 |
| **Best aggregated EER** | **0.062** | **0.258** | **0.014** |

**RQ3: Architectural Design Impact.** We evaluate the influence of architectural choices and input representations by comparing five categories of models: LLM-based (our SAFARI-LLM), State Space Models (RawBMamba), Graph Neural Networks (AASIST, RawGAT-ST), Convolutional Neural Networks (RawNet2, RawNet3), and Transformers (SSLModel, Conformer). We also analyze performance variations between raw waveform and spectrogram input representations.

**Implementation details and metrics:** All audio is resampled to 16 kHz mono. Spectrogram-based baselines use a 400-sample window with a 160-sample hop. LFCC features use 128 coefficients with $\Delta$ and $\Delta\Delta$; for some baselines, these cepstra are concatenated with Whisper embeddings. We use the Whisper-large variant throughout. For SAFARI–LLM, we adopt LLaMA-7B (Touvron et al., 2023) with LoRA adapters ($r$=8, $\alpha$=16). AFUM employs $K$=2 transformer projection experts (eight layers each; ∼88M parameters). Adapters operate at a fixed temporal stride of 80 ms, and the unified token dimensionality is $D$=2048. Since LLaMA-7B uses hidden size $d_{\text{LLM}}$=4096, we insert a learned input adapter $W_{\text{in}} \in \mathbb{R}^{2048 \times 4096}$ before the LLM. Models are trained with AdamW ($\beta_1$=0.9, $\beta_2$=0.95, weight decay 0.1). We report Accuracy and Equal Error Rate (EER).

**Parameter and efficiency accounting:** We *fine-tune* Whisper-large and m-HuBERT end-to-end, while keeping the LLaMA-7B backbone *frozen* except for LoRA adapters. Concretely, the trainable components are: (i) Whisper-large, (ii) m-HuBERT, (iii) AFUM, (iv) the stream adapters, (v) the 2048→4096 input adapter for LLaMA-7B, (vi) LoRA weights on the LLM, and (vii) the final MLP classifier. We show the parameter count of model components in Table 14.

**Reproducibility.** All the resources will be available at the project page.

Table 9: Cross-lingual deepfake detection results on the DECRO dataset between English and Chinese languages, demonstrating model performance when trained and evaluated across different language pairs.

| Models | Train on English, Eval on Chinese | | Train on English, Eval on English | | Train on Chinese, Eval on Chinese | | Train on Chinese, Eval on English | |
|---|---|---|---|---|---|---|---|---|
| | Acc | EER(%) | Acc | EER(%) | Acc | EER(%) | Acc | EER(%) |
| RawNet3 | 62.57 | 33.94 | 81.54 | 17.25 | 97.33 | 2.41 | 78.91 | 18.60 |
| Whisper-Mesonet | 66.30 | 21.82 | 72.09 | 15.91 | 66.43 | 7.78 | 72.25 | 21.27 |
| RawGAT-ST | 67.79 | 38.76 | 84.29 | 20.65 | 99.42 | 0.56 | 80.79 | 21.90 |
| AASIST | 68.29 | 32.49 | 84.56 | 16.52 | 98.47 | 1.17 | 82.22 | 9.53 |
| RawNet2 | 68.74 | 32.48 | 84.74 | 17.16 | 98.44 | 1.59 | **84.41** | 11.02 |
| Whisper-SpecRNet | 69.60 | 28.86 | 83.55 | 15.78 | 95.16 | 4.31 | 77.79 | 18.60 |
| Whisper-LCNN | 71.04 | 29.49 | 85.06 | 15.72 | 94.86 | 4.83 | 78.44 | 11.60 |
| RawBMamba | 71.49 | 29.17 | 86.12 | 14.93 | 98.33 | 1.56 | 81.90 | 17.60 |
| LCNN | 72.59 | 25.59 | 86.64 | 14.92 | 99.34 | 0.72 | 76.92 | 22.56 |
| Conformer | 72.86 | 42.89 | 86.55 | 22.38 | 98.23 | 1.52 | 76.54 | 20.66 |
| Whisper-frontend-LCNN | 80.65 | 25.83 | 90.57 | 14.62 | 98.20 | 0.96 | 77.36 | 8.72 |
| SpecRNet | 82.64 | 22.44 | 91.53 | 11.88 | 96.01 | 1.74 | 77.34 | 16.16 |
| SSLModel | 82.72 | 27.26 | 91.57 | 16.17 | 98.85 | 1.21 | 82.15 | 18.59 |
| MesoNet | 83.09 | 18.40 | 91.63 | 10.01 | 58.30 | 3.42 | 54.18 | 14.07 |
| **Proposed** | **84.48** | 21.20 | **92.44** | 11.35 | **99.60** | 0.36 | 82.70 | 11.80 |

## 5 Results and Analysis

This section presents a comprehensive evaluation of SAFARI-LLM's performance, addressing our three research questions related to cross-lingual generalization, extended cross-language family generalization, and the impact of model architecture. We report key metrics, compare SAFARI-LLM against baseline models, and embed detailed analyses and inferences within each subsection to elucidate trends and implications for multilingual deepfake detection.

### 5.1 Cross-Lingual Generalization Analysis (RQ1)

We tested SAFARI-LLM's cross-lingual generalization on the WaveFake and DECRO datasets. On Wave-Fake, SAFARI-LLM achieves near-perfect within-language detection: 99.99% accuracy (0.02% Equal Error Rate, EER) for English and 100% accuracy (0% EER) for Japanese, as shown in Table 6. On DECRO (Table 9), performance remains strong but reveals asymmetries, with 99.59% accuracy (0.36% EER) for Chinese compared to 92.43% accuracy (11.34% EER) for English. This discrepancy stems from dataset imbalances, with Chinese subsets having a real-to-fake ratio of 1:2 versus 1:3.4 for English, leading to higher false positives in English detection.

Cross-lingual evaluations highlight the challenges of language transfer. Training on the larger English Wave-Fake dataset (121,085 samples) yields robust generalization to Japanese (15,000 samples), achieving 94.21% accuracy (2.48% EER). Conversely, Japanese-to-English transfer results in 92.31% accuracy (5.31% EER), suggesting that larger, diverse training data enhances cross-lingual robustness.

We also evaluate SAFARI-LLM under the single-training-set protocol exactly as described in the Wave-Fake paper (Table 2). In this setting, the model is trained on one generator (row) and evaluated across all other generators (columns). The results for SAFARI-LLM in this protocol are shown in Table 7 and Table 8. SAFARI-LLM yields substantially lower aEERs for MelGAN, MB-MelGAN, and FB-MelGAN training, demonstrating stronger in-distribution and cross-vocoder generalization. FB-MelGAN training achieves an aEER of 0.014, compared to 0.062 (GMM) and 0.363 (RawNet2). These improvements suggest that SAFARI-LLM captures spoofing artifacts in a more transferable manner across vocoders, outperforming traditional GMMs and RawNet2 in most cases. SAFARI-LLM's performance is weaker on PWG and WaveGlow compared to GMM, which we attribute to vocoder-specific biases.

Table 10: Results with training models on Set A (Indo-European languages) and evaluating across Set A (within-family), Set B (Dravidian languages), and Set C (mixed languages) for Indic-Fake dataset, showing cross-language family generalization.

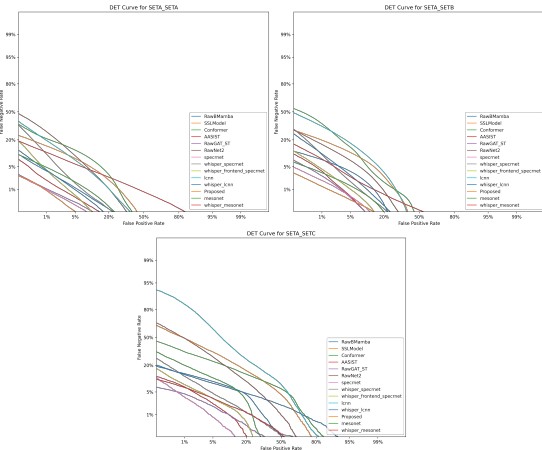

Figure 7: DET curves showing model performance when trained on Set A of the IndicFake dataset and evaluated on Set A, Set B, and Set C test sets.

| Models | Train on Set A, Eval on Set A | Train on Set A, Eval on Set B | Train on Set A, Eval on Set C |
|---|---|---|---|
| | EER (%) | EER (%) | EER (%) |
| RawBMamba | 2.217 | 4.155 | 7.800 |
| MesoNet | 2.746 | 2.244 | 8.545 |
| Whisper SpecRNet | 3.054 | 3.164 | 4.777 |
| Whisper MesoNet | 5.554 | 3.456 | 3.828 |
| LCNN | 6.151 | 9.968 | 22.750 |
| SSLModel | 6.742 | 8.213 | 17.544 |
| RawNet2 | 6.786 | 6.132 | 13.709 |
| Conformer | 8.881 | 8.177 | 15.335 |
| **Proposed** | **0.941** | **1.153** | **0.023** |

On DECRO, Chinese-to-English transfer yields 82.69% accuracy (11.79% EER), while English-to-Chinese achieves 84.48% accuracy but with a higher EER of 21.19%, indicating sensitivity to language-specific acoustic characteristics, particularly in prosodic and phonetic patterns.

SAFARI-LLM's dual-encoder architecture, combining Whisper's semantic embeddings and m-HuBERT's acoustic features, significantly outperforms single-stream models in cross-lingual settings. For instance, it surpasses RawNet3 by 19.98% in English-to-Japanese accuracy. However, the elevated EER in cross-lingual scenarios (e.g., 21.19% for English-to-Chinese) suggests residual sensitivity to language-specific acoustic artifacts. These results suggest that while semantic features facilitate robust generalization, acoustic variations across languages continue to pose a challenge. Future improvements should incorporate explicit phonetic modeling and balanced multilingual datasets to reduce false positives and enhance transferability, thereby ensuring the suitability of SAFARI-LLM for real-world multilingual deployment.

The higher EER for *English→Chinese* (21.20%) compared to *Chinese→English* (11.80%) on DECRO highlights residual language-specific artifacts and distributional mismatch. In contrast, within the Indic setting, where training and target languages are more closely aligned phonetically, cross-family EERs are much lower (e.g., **Set A→B: 1.15%**, **Set B→A: 3.78%**); and joint training on $A \cup B$ yields **0.72/0.73%** EER on Sets A/B and **0.68%** on unseen Set C. Together, these findings suggest that typological proximity and training coverage reduce score-distribution shift, while remaining errors motivate prosody-aware cues and light language-conditioned calibration.

## 5.2 Extended Cross-Language Family Generalization (RQ2)

We assessed SAFARI-LLM's generalization across language families using the IndicFake dataset, comprising Indo-European (Set A), Dravidian (Set B), and mixed languages (Set C). The results reveal several critical insights into cross-family transfer capabilities and architectural performance patterns.

### 5.2.1 Training on Set A (Indo-European Languages)

When training on Set A, SAFARI-LLM achieves strong in-family performance at 95.12% accuracy (0.94% EER) as shown in Table 10. This performance demonstrates excellent calibration, with the model achieving high accuracy while maintaining exceptionally low error rates. Notably, while baseline models like AASIST achieve higher accuracy (97.49%), their significantly higher EER (1.47%) indicates imbalanced class-specific performance, suggesting potential overfitting to the training distribution.

Table 11: Model performance when trained on Set B (Dravidian languages) and evaluated on Set A (Indo-European), Set B (within-family), and Set C (mixed languages) for Indic-Fake dataset, demonstrating cross-language family transfer capabilities.

| Models | Train on Set B, Eval on Set A | Train on Set B, Eval on Set B | Train on Set B, Eval on Set C |
|---|---|---|---|
| | EER (%) | EER (%) | EER (%) |
| LCNN | 4.507 | 3.215 | 25.597 |
| RawBMamba | 5.610 | 5.416 | 26.795 |
| Whisper SpecRNet | 5.676 | 4.175 | 19.298 |
| SSLModel | 7.162 | 5.801 | 18.246 |
| Whisper MesoNet | 7.966 | 4.821 | 7.689 |
| Conformer | 8.242 | 9.202 | 16.452 |
| RawNet2 | 8.921 | 9.496 | 24.740 |
| MesoNet | 14.568 | 9.331 | 30.524 |
| **Proposed** | **3.728** | **3.782** | **7.224** |

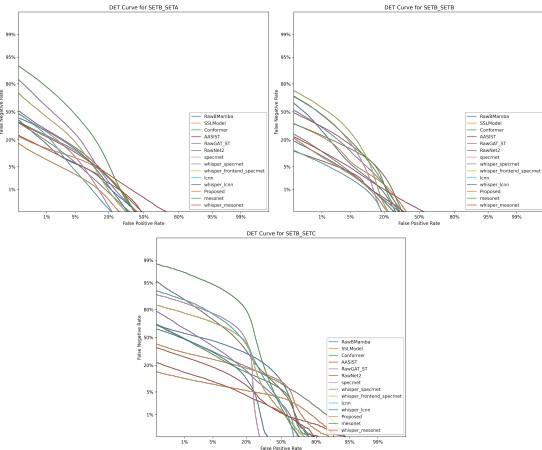

Figure 8: DET curves showing model performance when trained on Set B of the IndicFake dataset and evaluated on Set A, Set B, and Set C test sets.

The cross-family generalization results are particularly compelling. Testing the Set A-trained SAFARI-LLM on Set B yields 88.17% accuracy (1.15% EER), demonstrating robust cross-family transfer despite fundamental linguistic differences between Indo-European and Dravidian language families. This represents only a 6.95% accuracy drop with a minimal 0.21% EER increase, indicating excellent preservation of discriminative features across language families.

In contrast, other models show more dramatic performance degradation. For instance, Whisper MesoNet achieves higher cross-family accuracy (96.48%) but suffers from a substantially worse EER (3.45%), representing a 2.1x increase in error rate compared to SAFARI-LLM. This pattern suggests reduced reliability and potential overfitting to acoustic patterns specific to the training language family. We show the DET curve for each of the settings in Figure 7.

### 5.2.2 Training on Set B (Dravidian Languages)

Training on Set B reveals asymmetric transfer capabilities. SAFARI-LLM achieves 86.77% accuracy (3.78% EER) for in-family performance and maintains stable cross-family performance on Set A at 86.77% accuracy (3.72% EER). The remarkably consistent performance across both sets (86.77% accuracy) with nearly identical EER values (3.78% vs 3.72%) suggests that the model successfully learns language-agnostic features when trained on Dravidian languages.

However, a critical asymmetry emerges when comparing the effectiveness of Set A and Set B training. The Set A-trained model significantly outperforms the Set B-trained model on Set C (97.30% vs 60.69% accuracy), representing a 36.61% performance gap. This substantial difference indicates that Indo-European languages provide more transferable semantic and acoustic cues, likely due to their broader representation in pretrained foundation models like Whisper and m-HuBERT. We show the DET curve for each of the settings in Figure 8.

### 5.2.3 Joint Training Analysis

Joint training on Sets A and B (Table 12) achieves balanced performance: 83.92% accuracy (0.72% EER) on Set A and 84.64% accuracy (0.73% EER) on Set B. The near-identical EER values (0.72% vs 0.73%) and similar accuracy levels demonstrate successful knowledge integration across language families. This represents an 11.2% accuracy decrease from Set A-only training but achieves a much better balance, with only a 0.48% accuracy difference between families. Importantly, joint training dramatically improves Set C performance, achieving 60.45% accuracy (0.68% EER), which substantially outperforms Set B-only training

Table 12: Results from joint training on Set A and Set B, showing how combined training on Indo-European and Dravidian languages affects model performance across different language families.

| Models | Train on All, Eval on Set A | Train on All, Eval on Set B | Train on All, Eval on Set C |
|---|---|---|---|
| | EER (%) | EER (%) | EER (%) |
| LCNN | 1.232 | 0.892 | 6.496 |
| RawGAT-ST | 1.234 | 1.020 | 1.324 |
| AASIST | 1.306 | 0.726 | 3.110 |
| Whisper-Frontend-SpecRNet | 1.535 | 1.450 | 0.568 |
| RawBMamba | 1.897 | 3.224 | 6.358 |
| Whisper SpecRNet | 2.329 | 2.641 | 6.747 |
| SpecRNet | 2.597 | 1.483 | 9.921 |
| Whisper LCNN | 3.280 | 4.860 | 12.714 |
| Conformer | 3.344 | 2.029 | 10.828 |
| MesoNet | 3.948 | 2.902 | 14.409 |
| RawNet2 | 4.907 | 4.208 | 13.087 |
| SSLModel | 5.903 | 3.652 | 12.131 |
| Whisper MesoNet | 6.417 | 4.180 | 3.609 |
| **Proposed** | **0.725** | **0.729** | **0.680** |

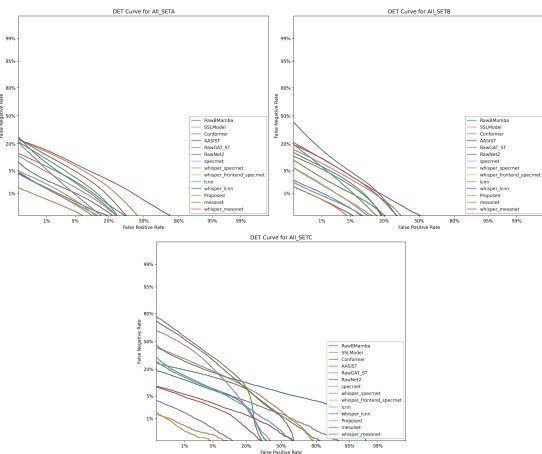

Figure 9: DET curves showing model performance when jointly trained on Set A and Set B of the IndicFake dataset and evaluated on Set A, Set B, and Set C test sets.

(60.69%) while maintaining the excellent calibration characteristics of SAFARI-LLM. We show the DET curve for each of the settings in Figure 9.

## 5.3 Impact of Model Architecture (RQ3)

We analyzed SAFARI-LLM's architectural contributions compared to baseline models. Raw-audio models like RawNet3 achieve near-perfect within-language accuracy (99.98%) but deteriorate sharply in cross-lingual settings (e.g., 75.23% accuracy for English-to-Japanese), indicating a strong dependency on language-specific acoustic features. Spectrogram-based models, such as Whisper-frontend-LCNN, show lower within-language accuracy but greater cross-lingual stability (85.09% English-to-Japanese, 80.48% Japanese-to-English), benefiting from language-agnostic pretrained embeddings.

SAFARI-LLM integrates the strengths of both approaches through its dual-stream architecture, leveraging Whisper for semantic features and m-HuBERT for acoustic cues. The Audio Feature Unification Module (AFUM) dynamically balances these representations, achieving a synergy that bridges cross-lingual gaps. For example, SAFARI-LLM outperforms RawNet3 by 19.98% and Whisper-frontend-LCNN by 9.12% in English-to-Japanese accuracy.

The LoRA-adapted LLaMA component in SAFARI-LLM strengthens multimodal integration by effectively aligning semantic and acoustic embeddings. This leads to consistent performance across languages and spoofing techniques. Importantly, the improvements appear to stem not from the linguistic pretraining of the LLM, but from its generalized ability to model dependencies across heterogeneous feature spaces.

This interpretation aligns with recent findings from LLM reprogramming literature ( Jin et al. (2024); Fan et al. (2024); Melnyk et al. (2023)), which view pretrained language models as universal sequence processors that can be adapted to new modalities through minimal parameter updates. SAFARI-LLM adheres to this principle: the LLM acts as a multimodal sequence integrator, facilitating the adaptive fusion of semantic and acoustic features while mitigating language-specific biases. Future work could explore more explicit reprogramming strategies and dynamic routing in AFUM to further enhance cross-domain generalization and resilience to emerging deepfake synthesis techniques.

## 5.4 Ablation Studies

To better understand the contributions of data scale and model design, we conduct two complementary ablations: (i) scaling training data within a language, and (ii) isolating the role of each feature stream and

Table 13: Within-language EER(%) as training samples increase. Gains differ by language family and representation in the pretraining pool.

| Training Samples | Hindi EER | Tamil EER |
|---|---|---|
| 10k | 3.91 | 4.23 |
| 20k | 2.35 | 3.74 |

Table 14: Ablation (EER %): contribution of the LLM and dual-stream fusion. *Base* retains both encoders and AFUM, replacing the LLM with a 2-layer MLP head to isolate the LLM's effect. All systems are trained on Set A ∪ Set B and evaluated individually on Set A, Set B, and Set C. We also show the parameter count for each of the configurations.

| Model Variant (Train = Set A∪Set B) | Parameter Count | Eval Set A | Eval Set B | Eval Set C |
|---|---|---|---|---|
| Base (dual-stream + AFUM, MLP head; no LLM) | 893,446,276 | 6.532 | 15.405 | 23.688 |
| Whisper + AFUM + LLM | 7,533,161,476 | 4.762 | 11.079 | 21.097 |
| m-HuBERT + AFUM + LLM | 6,983,075,204 | 5.818 | 13.362 | 21.970 |
| **SAFARI-LLM (Whisper + m-HuBERT + AFUM + LLM)** | 7,631,861,892 | **0.725** | **0.729** | **0.680** |

the LLM backbone. These controlled studies provide concrete evidence for the effectiveness of SAFARI-LLM.

### 5.4.1 Effect of Scaling Within a Language

We examine the effect of increasing training data for two typologically distinct Indic languages—Hindi (Indo–European) and Tamil (Dravidian). The evaluation is performed in a within-language setting, isolating the impact of additional data without cross-lingual transfer. We find that increasing Hindi data from 10k to 20k samples reduces EER by ∼40% (3.91 →2.35), while Tamil improves by only ∼12% (4.23 →3.74). This suggests that the marginal utility of additional data depends on the language family and its representation in the pretraining pool. Languages already well represented (e.g., Hindi) benefit more strongly from scaling, whereas typologically distant and under-represented languages (e.g., Tamil) remain challenging even with larger data sizes.

### 5.4.2 Effect of Model Components

We evaluate the contribution of each component using a controlled ablation with four settings trained on Set A∪Set B and evaluated on Set A, Set B, and Set C: (i) a *dual-stream* baseline that retains both encoders and AFUM but replaces the LLM with a 2-layer MLP head; (ii) Whisper + LLM (semantic stream only); (iii) m-HuBERT + LLM (acoustic stream only); and (iv) the full SAFARI-LLM with both streams unified by AFUM and processed by the LoRA-adapted LLM. This isolates the effect of the LLM from (a) dual-stream fusion and (b) AFUM.

Table 14 shows that adding an LLM to a *single* stream yields consistent but modest gains over the dual-stream MLP baseline, whereas the *largest improvement* occurs when **both** semantic and acoustic streams are fused by AFUM and integrated by the pretrained LLM (0.73/0.73/0.68 EER). This supports our hypothesis that *complementary* semantic+acoustic cues, integrated by a pretrained multilingual sequence model with long-range attention, are crucial for robust cross-family and zero-shot transfer (cf. Table 10).

## 6 Limitations and Future Work

While SAFARI-LLM advances multilingual deepfake detection, several limitations remain. First, our dataset, IndicFake, is region-focused and primarily targets Indic languages. We position it as a complement to existing

global resources rather than a replacement and therefore refrain from claiming universal generalizability from IndicFake alone. To contextualize the scope, we also evaluate the proposed methodology on non-Indic benchmarks, WaveFake (English ↔ Japanese) and DECRO (English ↔ Chinese), so that model relevance is assessed beyond the Indic region (see Table 6 and Table 7).

Second, the synthetic speech sources used in this work cover a few representative TTS architectures; Expanding to more diverse and stronger generators is an important next step. Third, although SAFARI-LLM achieves strong cross-lingual transfer, performance remains challenging between typologically distant languages (e.g., English → Chinese), highlighting persistent language-specific artifacts. Fourth, we do not explicitly evaluate robustness to partial manipulations or controlled noise/codec/channel effects, though the diversity of YouTube-sourced real data provides some natural robustness. We view these limitations as opportunities for future work: expanding coverage to additional language families, incorporating more diverse synthesis pipelines (including stronger contemporary generators), ensuring fully speaker-disjoint test splits, extending the framework to partial-fake detection and channel/codec robustness, and developing lighter-weight variants (e.g., distillation/compression) for edge deployment. In addition, exploring modern time-series foundation models such as Moment, TimesFM, or Granite TSPulse as alternatives to the LLM backbone presents a promising direction for reducing model size while retaining strong representational capacity. Finally, the architecture integrates Whisper-large, m-HuBERT, and a 7B LLM backbone; despite parameter-efficient LoRA fine-tuning, this remains computationally heavier than several baselines.

We view these limitations as opportunities for future work: expanding coverage to additional language families, incorporating more diverse synthesis pipelines (including stronger contemporary generators), ensuring fully speaker-disjoint test splits, extending the framework to partial-fake detection and channel/codec robustness, and developing lighter-weight variants (e.g., distillation/compression) for edge deployment. In addition, exploring modern time-series foundation models such as Moment Goswami et al. (2024), TimesFM Das et al. (2024), or Granite TSPulse Anonymous (2025) as alternatives to the LLM backbone presents a promising direction for reducing model size while retaining strong representational capacity.

## 7    Conclusion

This work introduces two transformative contributions to multilingual deepfake detection: the IndicFake dataset and the SAFARI-LLM model. The IndicFake dataset, encompassing over 4.2 million audio samples across 18 Indian languages from the Indo-European, Dravidian, and Sino-Tibetan families, establishes a new benchmark for linguistic diversity in deepfake research. IndicFake exhibits minimal overlap (Jaccard similarity ranging from 0.00 to 0.06) when compared individually to existing datasets, making it a robust resource for evaluating detection models across varied linguistic contexts. The proposed SAFARI-LLM, a novel dual-stream architecture, seamlessly integrates Whisper's semantic embeddings and m-HuBERT's acoustic features through an adaptive Audio Feature Unification Module (AFUM). Enhanced by a LoRA-fine-tuned LLaMA-7B model, SAFARI-LLM achieves state-of-the-art performance, delivering superior accuracy, exceptionally low error rates, and robust generalization across diverse languages and synthesis methods. Comprehensive experiments on IndicFake, DECRO, and WaveFake datasets demonstrate SAFARI-LLM's ability to balance semantic and acoustic information, outperforming existing models in cross-lingual and cross-language family scenarios while maintaining stability across varied deepfake generation techniques.

These advancements set a new standard for multilingual deepfake detection, offering scalable and reliable solutions for real-world deployment. In the future, we aim to expand IndicFake to include additional low-resource languages, further broadening its applicability. Optimization efforts will focus on model compression and efficient adaptation to enable deployment in resource-constrained environments. Additionally, integrating phonetic-aware modeling and targeted artifact identification will enhance cross-lingual robustness, paving the way for universally effective audio deepfake detection systems.

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
