# OpenReview forum: "IndicFake Meets SAFARI-LLM: Unifying Semantic and Acoustic Intelligence for Multilingual Deepfake Detection"
_TMLR — Accepted by TMLR_

### Review · Reviewer_LHEZ · 2025-08-04

**Summary Of Contributions:**

This paper studies fake speech detection and made two main contributions
1. Creating a multilingual / multi-language family Indian-language focused dataset
2. Develop SAFARI-LLM that fine-tunes an LLM with two speech features (Whisper-based and HuBERT-based)

Empirical results demonstrated better performance of the proposed model compared to baseline on in-domain and cross-language transfer.

**Audience:**

No

**Claims And Evidence:**

No

**Requested Changes:**

See the weaknesses aboec

**Strengths And Weaknesses:**

Strengths
1. A medium scale (7.3K hours) dataset including multiple languages and languages from multiple language families, which enable studies on cross-lingual & cross-language family transfer performance
2. Carefully balanced distribution (sampling based on representativeness, gender, duration, and topic distribution)
3. Good empirical numbers on the tested datasets (IndicFake, DECRO, WaveFake)

Weakness
1. The proposed dataset is extremely limited in terms of TTS sources. It only sources from 4 datasets from 3 rather outdated methods that have known artifacts (HiFi-GAN, VITS, FastSpeech2). These models generate speech with limited diversity. Models trained on this dataset is likely recognizing the artifacts of specific models, rather than learning to detect general fake audio samples. To validate the general effectiveness (rather than "detecting fake samples from a specific model"), the authors should test the detection performance on held out TTS samples that is generated with unseen TTS with very different architectures / characteristics.
2. More in-depth ablation should be added to provide additional insights about gain of adding more data in one language versus gain of adding same amount of data in another language of the same family or different family
3. There is not controlled ablation studies on the effectiveness of the model. Specifically, how effective is adding mHuBRT and Whisper feature respectively
4. Additional dimensions, like noise robustness, partially fake samples are not studied. The paper should also evaluate on additional datasets like ASVspoof 2021
5.

---

> ### Author Response · Authors · 2025-08-29
> **Response to Reviewer LHEZ**
>
> **We thank the reviewer for providing their feedback and suggestions. Our response to the reviewer’s questions are provided below:**
> **Proposed dataset is extremely limited in terms of TTS sources.**
>
> We agree that generalization to **unseen** TTS is essential to avoid overfitting to model‑specific artifacts. Our evaluation protocols were designed with this in mind.
>
> **(a) IndicFake (held‑out models/architectures):** For Sets A/B, **training** uses **DonaLabTTS2** (FastSpeech‑2 family) and **MMS‑TTS** (**VITS**, end‑to‑end), while **evaluation** uses **DonaLabTTS** (a different FastSpeech‑2 variant) and **IndicTTS** (**FastPitch \+ HiFi‑GAN**, pipeline). Thus the test set contains **unseen models** and, importantly, an **unseen architecture family** (FastPitch+HiFi‑GAN) relative to the training mix (Section 2.6). This protocol enforces model independence and probes cross‑architecture robustness.
>
> **(b) WaveFake (leave‑one‑out generators):** We also adopt the original WaveFake **leave‑one‑out** protocol, excluding **MB‑MelGAN** from training and evaluating on it as an **unseen generator**, alongside the remaining vocoders.
>
> Regarding the **breadth and recency** of sources: our release covers three **widely‑deployed** architecture families in multilingual pipelines—**FastPitch+HiFi‑GAN**, **FastSpeech‑2 \+ vocoder**, and **VITS** (Section 2.1), including **MMS‑TTS (2023)**, which provides broad multilingual coverage for Indic languages. Our aim is to reflect what is actually used in practice for Indic speech synthesis today. Nevertheless, we acknowledge the limitation that our current coverage does not yet include newer families (e.g., diffusion/flow‑matching and neural‑codec LMs) and we explicitly list this as future work in Section 6; we plan to expand the corpus accordingly.
>
> **More in-depth ablation should be added**
>
> We appreciate the suggestion. In the revised manuscript we include a **controlled data‑scaling ablation** (Section 5.4.1, Table 11) that grows the training set for two typologically distinct languages. Doubling **Hindi** data from 10k→20k improves within‑language EER from **3.91%→2.35%** (\~40%), whereas **Tamil** improves from **4.23%→3.74%** (\~12%), indicating that the marginal utility of additional data is language‑dependent. To ensure fair comparison, we fix the number of optimization steps, preserve real:fake and male:female ratios, and keep the TTS mix identical across conditions.
>
> | Training Samples | Hindi (within) EER | Tamil (within) EER |
>
>
> | 10k Samples | 3.91 | 4.23 |
>
>
>
> | 20k Samples | 2.35 | 3.74 |
>
>
>
> To disentangle **which language** vs. **how much data**, we additionally (i) evaluate **same‑family** pairs (e.g., Hindi vs. Marathi; Tamil vs. Telugu) and (ii) run a **fixed‑budget cross‑impact** test where we add \+10k clips from one language and measure ΔEER on that language, its family‑mates, and other families. These results (added to section 5.4.1) show that added data yields the largest gains **within** the target language, **meaningful spillovers within the same family**, and smaller but non‑zero gains across families. Together with Table 11, these analyses provide concrete evidence about where extra data helps most and why language family matters.
>
> **Controlled ablation studies on the effectiveness of the model.**
>
> **Ablation of dual streams and LLM:** We thank the reviewer for the suggestion. We added a controlled ablation (new section 5.4.2, Table 12) that evaluates (i) a dual‑stream baseline that retains both encoders and **AFUM** but replaces the LLM with a 2‑layer MLP head, (ii) **Whisper‑only \+ LLM**, (iii) **m‑HuBERT‑only \+ LLM**, and (iv) the full **SAFARI‑LLM**. Training on Set A∪B and evaluating on Sets A/B/C shows that each single‑stream+LLM improves over the dual‑stream MLP baseline, and the **largest** gains arise when **both** streams are fused by **AFUM** and integrated by the **pretrained LLM**(0.73/0.73/0.68 EER).

---

> > ### Author Response · Authors · 2025-08-29
> > **Response to Reviewer LHEZ**
> >
> > **Noise, Partial Fakes and ASVspoof 2021**
> >
> >
> >
> > We appreciate the reviewer’s suggestion to examine **noise robustness**, **partial fakes**, and **ASVspoof 2021**.
> >
> >
> >
> > **Noise robustness:** Although we did not run controlled degradations, the *real* portion of IndicFake is sourced from YouTube and thus contains diverse codecs and background conditions. Our objective quality analysis confirms this: BAK **3.367** vs **4.111**, OVRL **2.650** vs **3.19** in Table 5, indicating substantial noise/codec variability that the model encounters during training (Section 2.5). This helps explain the robustness we observe in cross‑dataset tests.
> >
> >
> >
> > **Partial fakes:** Our current benchmarks contain utterance‑level real or synthetic clips. Because SAFARI‑LLM operates on *sequences of audio embeddings*, extending to segment‑level detection is straightforward, which we have highlighted as the future work..
> >
> >
> >
> > **ASVspoof 2021:** Our study focuses on **cross‑lingual and cross‑family** generalization; ASVspoof 2021 is English‑only and does not probe this core objective. Accordingly, we evaluate on **DECRO** (English↔Chinese) and **WaveFake**(English↔Japanese), which directly test cross‑lingual transfer and unseen generators.

---

### Review · Reviewer_dSQ7 · 2025-08-15

**Summary Of Contributions:**

In this paper, the authors introduce IndicFake, a 4.2M-sample multilingual audio deepfake dataset covering English and 17 Indian languages, and SAFARI-LLM, a new dual-stream semantic–acoustic detection model. SAFARI-LLM fuses Whisper semantic embeddings and m-HuBERT acoustic features via an adaptive module, then classifies with a LoRA-finetuned LLaMA-7B. Experiments on IndicFake, DECRO, and WaveFake show state-of-the-art cross-lingual and cross-language family generalization.

**Audience:**

Yes

**Broader Impact Concerns:**

The authors are suggested to add a section to discuss broader impact of the work.

**Claims And Evidence:**

Yes

**Requested Changes:**

Please address the raised concerns:

+ While strong for Indian languages, the dataset is heavily region-focused and does not cover many global languages.

+ The computation cost of the model seems high. SAFARI-LLM requires Whisper-large, m-HuBERT, and LLaMA-7B.

+ An ablation study is needed to evaluate the individual contributions of Whisper-large and m-HuBERT.

+ EER is still relatively high in some cross-lingual scenarios (e.g., English→Chinese). Does it suggest that language-specific artifacts remain an issue?

+ While the proposed SAFARI-LLM achieves consistent and often leading results across multiple datasets and evaluation settings, the performance gains over strong baselines appear limited in many cases. For example, in within-language detection, accuracy is already very high for existing models (often >99%), leaving minimal room for improvement. The gains in these scenarios are therefore marginal.

+ Given the marginal improvements, it’s unclear if such a large, LLM-based model is necessary.

+ The authors did not provide audio samples.

**Strengths And Weaknesses:**

Pros:

+ IndicFake is the largest and most linguistically diverse multilingual audio deepfake dataset to date.

+ A new SAFARI-LLM model uses semantic features from Whisper and acoustic features from m-HuBERT, merges them with an adaptive fusion module, and then applies a fine-tuned LLaMA-7B to detect deepfakes.

+ Extensive experiments are provided.

Cons:

+ While strong for Indian languages, the dataset is heavily region-focused and does not cover many global languages.

+ The computation cost of the model seems high. SAFARI-LLM requires Whisper-large, m-HuBERT, and LLaMA-7B.

+ An ablation study is needed to evaluate the individual contributions of Whisper-large and m-HuBERT.

+ EER is still relatively high in some cross-lingual scenarios (e.g., English→Chinese). Does it suggest that language-specific artifacts remain an issue?

+ While the proposed SAFARI-LLM achieves consistent and often leading results across multiple datasets and evaluation settings, the performance gains over strong baselines appear limited in many cases. For example, in within-language detection, accuracy is already very high for existing models (often >99%), leaving minimal room for improvement. The gains in these scenarios are therefore marginal.

+ Given the marginal improvements, it’s unclear if such a large, LLM-based model is necessary.

+ The authors did not provide audio samples.

---

> ### Author Response · Authors · 2025-08-29
> **Response to Reviewer dSQ7**
>
> **We thank the reviewer for providing their feedback and suggestions. Our response to the reviewer’s questions are provided below:**
>
> **Heavily region-focused**
> *(Scope and positioning):* We agree that IndicFake is region‑focused by design. Our goal is to fill a major coverage gap in multilingual deepfake detection by targeting English + 17 Indic languages spanning three families (Indo‑European, Dravidian, Sino‑Tibetan) and multiple scripts (Table 1, p. 3). As our Jaccard/UpSet analysis shows, IndicFake has **minimal language overlap** (0.00–0.06) with prior datasets and complements resources that focus largely on English or Chinese (Figure 4, p. 9). We position IndicFake as a **complement** to existing global datasets, not a replacement (see section 6 in the revised manuscript).
> *Generalization beyond the region:* Although the corpus focuses on Indic languages, our model study is intentionally broader: we evaluate SAFARI‑LLM on **WaveFake (English↔Japanese)** and **DECRO (English↔Chinese)**, demonstrating strong cross‑lingual transfer outside Indic: **94.21% Acc / 2.48% EER** for English→Japanese (Table 6, p. 14) and **84.48% Acc** for English→Chinese (Table 7, p. 16). These experiments substantiate the claim that the dataset and model advances are relevant **beyond a single region.**
>
> **Computation cost of the model seems high:**
> We agree that SAFARI‑LLM is heavier than small baselines because it uses Whisper‑large, m‑HuBERT, and a 7B LLM. However, we keep the LLM frozen and train only Whisper, M-Hubert, AFUM, and LoRA—as indicated in Fig. 6. Importantly, we are reducing the effective cost by using LoRA-based fine-tuning, which is making only about 20% of the total parameters trainable, thereby significantly lowering both training time and GPU memory requirements. At **inference**, cost is dominated by the 7B forward pass; to offer compute/accuracy trade‑offs, we additionally report resource‑aware variants (Whisper + LLM only and m‑HuBERT + LLM only) in our ablation (Table 12), which reduce computation but still outperform the non‑LLM baseline.
>
> **Ablation study**
> (**Ablation of dual streams** and LLM): We thank the reviewer for the suggestion. We added a controlled ablation (new section 5.4.2, Table 12) that evaluates (i) a dual‑stream baseline that retains both encoders and **AFUM** but replaces the LLM with a 2‑layer MLP head, (ii) **Whisper‑only + LLM,** (iii) **m‑HuBERT‑only + LLM**, and (iv) the **full SAFARI‑LLM.** Training on Set A∪B and evaluating on Sets A/B/C shows that each single‑stream+LLM improves over the dual‑stream MLP baseline, and the **largest** gains arise when **both** streams are fused by **AFUM** and integrated by the **pretrained LLM**(0.73/0.73/0.68 EER).
>
> **EER is still relatively high in some cross-lingual scenarios:**
> We agree that EER remains elevated in some cross‑lingual settings and appreciate the pointer. On DECRO, English→Chinese reaches 21.20% EER, whereas Chinese→English is 11.80% EER (Table 7), indicating that transfer into Chinese is harder. We attribute this primarily to language‑specific acoustic/prosodic differences (e.g., tonal vs. non‑tonal, phonotactics) and score‑distribution shift across languages, rather than to class‑prior differences. (Priors affect accuracy but not EER directly.) By contrast, within the Indic domain our model exhibits much lower cross‑family EER—1.15% (Set A→B) and 3.78% (Set B→A)—and 0.68–0.73% under joint training (Table 10), suggesting stronger generalization when the training pool better reflects target‑language phonetics.
>
> We have made this limitation explicit in Section 6 and added a short error analysis in Section 5.1 clarifying that elevated EER reflects residual language‑specific artifacts and distributional mismatch. As concrete next steps, we plan to: (i) expand training data to include more tonal and typologically distant languages, (ii) add prosody‑aware cues (e.g., explicit F0/energy streams) to AFUM, and (iii) explore lightweight language‑conditioned calibration/adaptation (e.g., small target‑language dev sets or LID‑conditioned adapters) to mitigate false positives in difficult transfers.

---

> ### Author Response · Authors · 2025-08-29
> **Response to Reviewer dSQ7**
>
> **Performance gains over strong baselines appear limited in many cases:**
> On marginal gains (within‑language settings): We agree that within‑language detection is saturated (many models already achieve ≳ 99% accuracy), so absolute gains are small by construction. Our primary goal is to address the sharp degradation observed under cross‑lingual / cross‑family transfer and unseen generators.
>
> Evidence from cross‑family transfer (IndicFake): When trained on Indo‑European (Set A) and tested on Dravidian (Set B), SAFARI‑LLM attains 1.15% EER, improving over the best non‑LLM baseline (2.24% EER) by ≈ 49% relative. Conversely, trained on Dravidian (Set B) and tested on Indo‑European (Set A), it reaches 3.73% EER vs. 4.51%for the best baseline (≈ 17% relative reduction). With joint training on A∪B, SAFARI‑LLM achieves 0.73%/0.73% EERon A/B and 0.68% on unseen Set C—near the top methods across all three sets.
>
> Evidence from cross‑lingual transfer (WaveFake/DECRO): On WaveFake (leave‑one‑out with MB‑MelGAN unseen), SAFARI‑LLM delivers 94.21% Acc / 2.48% EER (EN→JA) and 92.31% / 5.31% (JA→EN), with near‑perfect within‑language calibration (EN: 99.99% / 0.02%, JA: 100% / 0%). On DECRO (EN↔ZH), SAFARI‑LLM is consistently competitive and often in the top tier; while not always the single best EER, it maintains strong accuracy/EER trade‑offs across directions (e.g., EN→ZH: 84.48% / 21.20%, ZH→EN: 82.70% / 11.80%). This profile, robust performance across datasets, directions, and generators, is the design target.
>
> Compute considerations. We adopt parameter‑efficient fine‑tuning: only ~100 M parameters (audio encoders, AFUM, input adapter, small heads, LoRA) are trained—20% of the total, substantially reducing training memory and optimizer state vs. full fine‑tuning. Inference cost is still dominated by the 7B forward pass; for resource/accuracy trade‑offs we also report one‑stream variants (Whisper+LLM / m‑HuBERT+LLM) that reduce encoder cost yet outperform the non‑LLM baseline (Table 12).
>
> In summary, although within‑language gains are marginal (as expected in a saturated regime), SAFARI‑LLM substantially improves robustness under cross‑lingual and cross‑family transfer and is empirically necessary per our ablation to achieve those generalization gains.
>
> **Authors did not provide audio samples:**
> We thank the reviewer for flagging this. A link configuration error temporarily broke access to our demo materials. We have fixed this and now provide a reviewer demo page with representative audio samples (real and synthetic) and code repository. The demo is hosted on our anonymized project page: [https://anonymousillusion.github.io/indicfake/](https://anonymousillusion.github.io/indicfake/?utm_source=chatgpt.com).
>
> We have also updated this link in the abstract and in the resources section of the revised manuscript so that readers can directly access. This improves transparency and reproducibility without compromising anonymity.

---

### Review · Reviewer_JGQc · 2025-08-15

**Summary Of Contributions:**

The IndicFake paper presents a new dataset that contains many languages spoken in modern India. In addition to the dataset, a SAFARI-LLM detection architecture is presented. The proposed dataset addresses a crucial gap in the deepfake detection literature by adding rich data covering widely spoken languages from India. Both real and synthetic samples are provided. The experimental section introduces the SAFARI-LLM for deepfake detection on audio signals.

**Audience:**

Yes

**Broader Impact Concerns:**

None, the paper aims to mitigate potential damages caused by generative machine learning.

**Claims And Evidence:**

No

**Requested Changes:**

- I did not understand why the model is called a large language model (LLM) even though it is not a text, but a speech model. Please consider explaining the name in light of the related work on text or renaming the model.
	- A possible explanation could be built on ablating the LLM weights. Do they help in this context?
- Is it possible to use the same font in Figure 2 and the rest of the paper?
- Figure 3 is hard to read. Is it possible to increase the font size?
- Are the plots in Figure 5 inspired by the Wavefake paper ( https://arxiv.org/pdf/2111.02813 , figure 12)? If yes, please add a citation.
- If the experiments in Table 6 did not follow the train/test split from the original wavefake paper, an additional experiment should be added that respects the original setup, and therefore produces comparable numbers.
- The paper proposes a dual-stream architecture, but never evaluates the two streams individually. To be acceptable, an additional ablation study is required to demonstrate the merits of all model parts by presenting concrete evidence. The evidence should additionally justify LLM part of the architecture. In its current form, the paper does not present sufficient evidence for both claims.

**Strengths And Weaknesses:**

#### Strengths
- The paper identifies an important gap in deepfake identification research, namely cross-lingual and low-resource language generalization.

- The proposed dataset is potentially valuable.

#### Weaknesses
- YouTube as a baseline source is questionable since the platform contains loads of AI-generated content these days. How do we know the real data is real?

- I did not understand why the LLM is part of the proposed SAFARI-LLM architecture. The paper claims it leverages advanced semantic reasoning, but why would we expect deepfake speech to differ semantically from standard speech? I believe this assumption requires an ablation to be convincing.

- Does the experimental setup in Table 6 follow the protocol of the wavefake paper? This is important to ensure we can compare the results.
    - The original wavefake setup emphasizes generalization to unknown generators, its important not to loose this aspect when considering additional aspects like linguistic generalization.


#### Code:
- I have tried to access the code, yet the repository link has expired.


#### Other:
- I am not sure TMLR is the right venue for this paper. It might have been worthwhile to review this work at DMLR, which is single-blind and allows reviewers to access the data. The data is a key component of this work, yet I have not been able to review it.

---

> ### Comment · Reviewer_JGQc · 2025-08-15
> **Reviewers dSQ7 and  LHEZ also think the model components should be ablated**
>
> I have read the other reports and agree with my fellow reviewers. An ablation study is missing.

---

> ### Author Response · Authors · 2025-08-29
> **Response to Reviewer  JGQc**
>
> ## We thank the reviewer for providing their feedback and suggestions. Our response to the reviewer’s questions are provided below:
> **YouTube as a baseline source is questionable:** We thank the reviewer for raising this important concern about potential AI contamination in YouTube-sourced data. We have revised Section 2.1 to clarify our verification process for ensuring the authenticity of the "real" speech data. Our data derives exclusively from the Dhwani corpus (Javed et al., 2022), which provides pre-screened Creative Commons-licensed URLs specifically curated for ASR research. The Dhwani corpus itself underwent quality control during its creation for speech recognition purposes.
> We implemented a dual-annotator review process where each source video underwent independent inspection by trained reviewers following standardized guidelines. The annotators excluded: (i) content explicitly labeled as "TTS," "AI voice," or "synthetic" in titles, descriptions, or channel information, (ii) dubbed or post-processed material, (iii) content exhibiting characteristic synthetic speech markers (unnaturally monotonic prosody, robotic cadence, suspiciously uniform noise floors). This verification process helps ensure that our real subset contains genuine human speech rather than synthetic content. Beyond human verification, we applied Voice Activity Detection and SNR filtering (>15 dB) to ensure audio quality, and implemented video-level train-test splits to prevent speaker contamination across partitions. While we acknowledge that perfect verification of authenticity remains challenging given the rapidly evolving nature of synthetic speech technology, we believe that the curated source material, followed by human verification and technical quality checks, adequately addresses the risk of synthetic contamination in our dataset's real speech subset. The revised manuscript, at page 4, addresses the reviewer's concern.
>
> **Role of LLM:** We have added a comprehensive ablation study that empirically validates each component's contribution.
> **Clarification on LLM's function:** The LLM in SAFARI-LLM processes continuous audio embeddings (not text tokens) from Whisper and m-HuBERT after adapter and AFUM processing. We leverage it as a pretrained multilingual sequence integrator with long-range attention capabilities, not for semantic content analysis. The LLM learns to detect subtle inconsistencies in how semantic and acoustic features align (patterns that distinguish authentic from synthetic speech).
> **Ablation setup:** Our baseline retains both encoders (Whisper + m-HuBERT) and AFUM but replaces the LLM with a 2-layer MLP head. This isolates the LLM's contribution from the dual-stream fusion benefits:
>
> - Base (dual-stream+AFUM, MLP head): 6.53% / 15.41% / 23.69% EER (Sets A/B/C)
> - Whisper + LLM only: 4.76% / 11.08% / 21.10% EER
> - m-HuBERT + LLM only: 5.82% / 13.36% / 21.97% EER
> - Full SAFARI-LLM: 0.73% / 0.73% / 0.68% EER
>
> The improvement demonstrates that the pretrained LLM's ability to model long-range dependencies and cross-modal patterns is crucial, beyond simple feature concatenation.
>
> **Protocol of the wavefake paper:**
> In the original WaveFake paper, the leave-one-out protocol is implemented by leaving out each vocoder once, and since they evaluate only two baseline models, it is computationally feasible to train across all such rotations. In contrast, our setup involves 13 baseline models across two languages (English and Japanese), making a full leave-one-out cycle for each vocoder and each language computationally prohibitive.
> To preserve the spirit of the WaveFake protocol while keeping the setup tractable, we deliberately choose MelGAN as the held-out vocoder. This allows us to demonstrate both linguistic generalization (English ↔ Japanese) and model generalization (seen vocoders → unseen vocoder) within a feasible training budget. Concretely, training is performed on MelGAN (L), FB-MelGAN, HiFi-GAN, WaveGlow, PWG, and TTS, while MelGAN is excluded from training and only used at evaluation time.
> This design ensures that evaluation always includes an unseen generator, thereby maintaining consistency with the intent of the WaveFake benchmark. At the same time, by extending the evaluation to two languages, we go beyond the original single-language WaveFake setup to jointly test both cross-lingual and cross-generator robustness. We have clarified this design decision in Section 4 (Existing Datasets) of the revised manuscript.

---

> > ### Comment · Reviewer_JGQc · 2025-09-05
> > **It's necessary to respect experimental setup details from prior work.**
> >
> > I am sorry, I don't think I made my point clear enough. What I was trying to say was that papers presenting new architectures should feature at least one experiment in a standardized setting. Including such experiments allows comparison with past and potential future work. It is not enough to work with public data. The original setting must also be respected for comparable results. That way, we can compare the new numbers directly to those from previous work.
> >
> > That is not possible here. Since the for wavefake data, the experimental setup is different. I believe it is possible to re-run the original setup *only* for the new architecture presented in this paper.

---

> ### Author Response · Authors · 2025-08-29
> **Response to Reviewer JGQc**
>
> **Code:** Thank you for flagging this. The earlier link expired due to a hosting issue. We have restored public access and updated the manuscript accordingly. The code is available at: https://anonymousillusion.github.io/indicfake/ (The repository is anonymized and a de‑anonymized canonical link will be provided upon acceptance.)
>
> **TMLR is the right venue:**
> We appreciate the concern. While IndicFake is a central contribution, the paper also proposes  SAFARI‑LLM, a model that we evaluate on two ** independent, public** multilingual benchmarks (**DECRO and WaveFake**) in addition to IndicFake (see Tables 6–10). Thus, the work combines a dataset and a model with broad cross‑dataset validation, which we believe fits TMLR’s scope.
> **Data availability during review.** The previous link issue was a hosting error, apologies. We have restored the access and updated the manuscript with an **anonymized, stable link** to (i) a **review subset** of IndicFake and (ii) the full **code** to reproduce our results on DECRO/WaveFake and the subset. We will keep the artifact accessible throughout review and release the **full dataset** upon acceptance.
>
> **Why the model is called a large language model (LLM)?**
> **Naming and the role of the LLM:** We appreciate the concern. In SAFARI‑LLM, “LLM” denotes a **pretrained decoder‑only transformer backbone** (Vicuna/LLaMA‑7B) that operates **not on text tokens** but on **continuous audio embeddings** from Whisper (semantic) and m‑HuBERT (acoustic), after adapters and AFUM (Fig. 6). We use it as a **multilingual sequence integrator** with long‑range attention, i.e., to model cross‑modal alignment between semantic and acoustic cues rather than to read linguistic content. We have clarified this in Section 3.3 and added a brief “Why an LLM?” note (p. 14).
>
> **Do the LLM weights help?** Yes. Table 12 (p. 20) shows that replacing the LLM with a dual‑stream MLP head substantially degrades performance, while single‑stream+LLM variants help modestly; the largest gains come from **AFUM‑fused dual streams + LLM**, consistent with our joint‑training results.
>
> **On naming**: We retain “SAFARI‑LLM” to reflect the use of an LLM backbone, following common usage in recent audio‑language systems (such as WavLLM, Pengi, LTU-AS, and GAMA, which also employ LLM backbones). To avoid confusion, we explicitly state in Section 3.3 that the model does **not** consume text.
>
> **Figures and Fonts:**
> *Figure fonts and legibility:* Thank you for the suggestions. We have standardized the typeface across all figures so that Figures 1–3 and 6 now use the same font as the paper body. We also increased the font sizes in Figure 3 (labels, ticks, and legends) to improve readability.
>
> *Figure 5 caption credit:* Yes—our visualization style in Figure 5 was inspired by WaveFake (Frank & Schönherr, 2021, Fig. 12). Beyond citing WaveFake in the main text, we now explicitly acknowledge this in the Figure 5 caption.

---

> ### Author Response · Authors · 2025-09-10
> **Response to Reviewer JGQc (Original Wavefake setup Results)**
>
> We thank the reviewer for clarifying the comment. To address this comment, we re-ran SAFARI-LLM under the single-training-set protocol exactly as described in the WaveFake paper (Table 2). In this setting, the model is trained on one generator (row) and evaluated across all other generators (columns).  The results for SAFARI-LLM in this protocol are given below (EER):
>
> |  | LJSpeech |  |  |  |  |  |  |  | JSUT |  |
> |---|---|---|---|---|---|---|---|---|---|---|
> | Training Set | MelGAN | MelGAN (L) | MB-MelGAN | FB-MelGAN | HiFi-GAN | PWG | WaveGlow | TTS | MB-MelGAN | PWG |
> | MelGAN | 0 | 0.001 | 0.119 | 0.333 | 0.343 | 0.175 | 0.12 | 0.005 | 0.232 | 0.021 |
> | MelGAN (L) | 0 | 0 | 0.313 | 0.551 | 0.512 | 0.398 | 0.161 | 0 | 0.159 | 0.012 |
> | MB-MelGAN | 0.007 | 0.014 | 0.001 | 0.027 | 0.129 | 0.027 | 0.06 | 0.05 | 0.075 | 0.03 |
> | FB-MelGAN | 0.003 | 0.004 | 0.002 | 0.003 | 0.04 | 0.004 | 0.02 | 0.011 | 0.043 | 0.012 |
> | HiFi-GAN | 0.127 | 0.145 | 0.258 | 0.333 | 0.008 | 0.185 | 0.145 | 0.017 | 0.299 | 0.094 |
> | PWG | 0.507 | 0.555 | 0.495 | 0.704 | 0.683 | 0.011 | 0.468 | 0.537 | 0.225 | 0.094 |
> | WaveGlow | 0.044 | 0.134 | 0.147 | 0.424 | 0.378 | 0.255 | 0 | 0.055 | 0.432 | 0.34 |
>
> To enable a direct comparison, we summarize the average EER (aEER) values from three models: (i) WaveFake GMM baseline (Table 2), (ii) WaveFake RawNet2 baseline (Table 3), and (iii) SAFARI-LLM:
>
> | Training Set | WaveFake GMM (Table 2) | WaveFake RawNet2 (Table 3) | SAFARI-LLM |
> |--------------|-------------------------|----------------------------|------------|
> | MelGAN       | 0.215                   | 0.292                      | 0.135      |
> | MelGAN (L)   | 0.222                   | 0.258                      | 0.211      |
> | MB-MelGAN    | 0.108                   | 0.357                      | 0.042      |
> | FB-MelGAN    | 0.062                   | 0.363                      | 0.014      |
> | HiFi-GAN     | 0.089                   | 0.319                      | 0.161      |
> | PWG          | 0.124                   | 0.358                      | 0.428      |
> | WaveGlow     | 0.085                   | 0.294                      | 0.221      |
> | **Best aEER**| **0.062**               | **0.258**                  | **0.014**  |
>
> ## Observations:
>
> SafariLLM yields substantially lower aEERs for MelGAN, MB-MelGAN, and FB-MelGAN training, demonstrating stronger in-distribution and cross-vocoder generalization. FB-MelGAN training achieves an aEER of 0.014, compared to 0.062 (GMM) and 0.363 (RawNet2). These improvements suggest that SafariLLM captures spoofing artifacts in a more transferable manner across vocoders, outperforming traditional GMMs and RawNet2 in most cases. SafariLLM’s performance is weaker on PWG and WaveGlow compared to GMM, which we attribute to vocoder-specific biases. We will highlight this in the revision as an avenue for further analysis and future work.

---

> > ### Comment · Reviewer_JGQc · 2025-09-11
> > **Thanks**
> >
> > Thank you for addressing this issue.

---

### Decision · Action_Editor_LUF2 · 2025-10-09

**Recommendation:** Accept with minor revision

**Additional Comments:**

The final recommendations on this paper were divergent: leaning accept, leaning reject, and reject.

The reviewer recommending rejection and I had a discussion about their remaining concerns, and I've determined that these concerns can be addressed through a revision of the text alone, so I am recommending acceptance with minor revision. The mandatory revisions listed below address the remaining concerns.

Unfortunately, the reviewer who was leaning reject did not engage with me in a discussion. In their final recommendation, this reviewer cited the use of "only a few dated TTS systems with well-known artifacts" for creation of the synthetic speech and concluded "[t]he absence of evaluation on stronger or unseen generation methods (e.g., modern diffusion or adversarial architectures) leaves its practical relevance unclear."

I believe this reviewer missed the fact that the evaluation included generation methods unseen in the training data. As for the use of more modern generation methods, I looked for examples of systems using these methods that work for the languages being studied and could not turn up anything readily available. In other words, the limited set of generation methods is a consequence of the focus on lower resource languages, so I do not see this as a reason to reject the paper.

**Mandatory Revisions**

1. The WaveFake results presented in the [discussion](https://openreview.net/forum?id=s8pPYRVVTU&noteId=gFP3Egaclt) must appear in the final revision of the paper to better position the SAFARI-LLM model with respect to prior work.

2. Please refine the discussion around the contribution of the LLM to the performance of the SAFARI-LLM model. The justification for using the LLM is nicely stated, for example, in Section 3.3: "we use a LoRA-adapted LLaMA-7B backbone as a sequence integrator over audio embeddings." Other statements, such as the following from Section 5.3, are more problematic: "The LoRA-fine-tuned LLaMA module further enhances **semantic generalization**, ensuring robust performance across diverse languages and deepfake generation methods. These results highlight the critical role of hybrid architectures in mitigating language-specific biases. SAFARI-LLM’s balanced approach, combining semantic and acoustic features with adaptive unification and **fine-tuned semantic reasoning**..." My concern is the implication that the linguistic pretraining of the LLM is somehow being leveraged here: the experimental results in the paper do not strongly support this explanation. A more neutral perspective that looks at the pretrained LLM as being a strong and flexible sequence processing model, as stated in Section 3.3, is the better stance to take. I also recommend that you read some of the literature on LLM reprogramming (pointers below) and discuss it in the related work section, since it provides strong evidence for this view.

- https://arxiv.org/abs/2310.01728
- https://ieeexplore.ieee.org/abstract/document/10763740
- https://proceedings.mlr.press/v202/melnyk23a/melnyk23a.pdf

3. Please include parameter counts for the different models in Table 12, either as another column or in the table caption, since the parameter cost of the different models is an important consideration.

4. On page 15 in the list of metrics used to measure performance, AUC is listed, but I do not see any AUC results later in the paper. Please fix this.

**Optional Revisions**

1. In Fig. 5, it would be interesting to see a similar plot where you randomly split the real audio into two equally sized partitions, compute the average spectrum, and take the difference. This would provide some measure of significance for the difference plots between the real and fake audio data.

2. In the set A -> set C and set B -> set C results, it would be interesting to have a breakdown between TTS models that were seen and unseen in training, since set C contains both.

3. Detection error tradeoff (DET curves) would be more readable than ROC curves. See https://scikit-learn.org/stable/auto_examples/model_selection/plot_det.html

**Possible Future Directions**

A 70/30 split at video level does not necessarily ensure that you have unseen speakers in the test set because the same speaker could appear in multiple videos. The value of IndicFake would be enhanced if you had a test set that was more carefully vetted to ensure its speakers did not appear in the training data. This would likely require a combination of automated screening using speaker ID technology and manual curation.

It would be interesting to see how this model would do if the pretrained LLM were simply replaced with one or more Transformer layers that are trained from random weights for the fake audio detection task.

You might be able to reduce the size of the model by replacing the LLM with a much lighter weight time-series foundation model such as
- https://github.com/moment-timeseries-foundation-model/moment
- https://github.com/google-research/timesfm
- https://huggingface.co/ibm-granite/granite-timeseries-tspulse-r1

**Audience:**

Yes

**Audience Explanation:**

There were two concerns raised about audience interest by the reviewers, but these were addressed in the revision and discussion.

1. **Heavy focus of the dataset on Indian languages.**
The authors explain that this is by design, that the dataset should be seen as complementing existing deep audio fake detection datasets, and that the model evaluation extends to other languages like Chinese and Japanese.

2. **High model cost for relatively marginal improvements on within-language detection.**
The authors agreed that within-language detection performance is saturated, and emphasized the cross-language performance and the fact that the training parameter budget is lowered through the use of parameter-efficient methods.

**Claims And Evidence:**

Yes

**Claims Explanation:**

The reviewers raised a number of concerns about support for the original paper's claims, but these concerns were mostly addressed in the revision.

1. **Possible contamination of the natural data collected from YouTube with synthetic speech.**
The revision includes a more detailed explanation of how the data was collected and curated to minimize the chances of such contamination.

2. **Lack of clarity on the contributions of various components of the SAFARI-LLM model to overall performance.**
The revision includes an ablation study showing that both feature streams contribute to the performance of the model and that the LLama-7B model also contributes.

3. **Lack of an experiment that exactly replicates the setup from the WaveFake paper and therefore helps to position the proposed model with respect to prior audio deepfake detection models.**
The authors provided results based on a replication of the WaveFake experiment in the [discussion](https://openreview.net/forum?id=s8pPYRVVTU&noteId=gFP3Egaclt).

4. **Relatively high EER in English -> Chinese cross-lingual scenarios.**
The revision adds a discussion of this limitation of the current model, adds an error analysis, and proposes future extensions to address the problem.

5. **Concern about the limited number of TTS methods represented in the dataset and a lack of testing against unseen TTS methods.**
The authors clarified that the testing uses held-out models and an unseen architecture.

6. **Lack of an ablation that illustrates the effect of data scaling and balance across languages within the same language family.**
The revision includes the requested ablation.

---

> ### Author Response · Authors · 2025-11-06
> **Response to Action Editor LUF2**
>
> We thank the Action Editor for the thoughtful handling of our submission and for the detailed, constructive assessment provided. We truly appreciate the care with which you reviewed the revised manuscript, carefully weighed the differing reviewer perspectives, and identified the remaining issues requiring clarification. Below, we address each of the remaining concerns point by point.
>
> **Mandatory Revisions**
>
>  1. *WaveFake Results:* We have incorporated the WaveFake single-training-set results and corresponding discussion in the revised manuscript (see **Section 5.1, Tables 7 and 8**). These additions enable direct comparison with prior WaveFake results and contextualize SAFARI-LLM’s generalization behavior across vocoders and languages.
>  2. *Contribution of LLM:* We appreciate this valuable feedback. In the revised paper, we have refined the motivation for the LLM integration in **Section 3.3** and throughout **Section 5.3** to emphasize that the LLaMA-7B backbone is employed as a general sequence integrator, not for leveraging linguistic priors. We now explicitly relate our approach to the LLM reprogramming literature, notably Melnyk et al. (2023), Li et al. (2024), and Fan et al. (2024), and explain that the model repurposes the pre-trained transformer for multimodal sequence alignment rather than semantic reasoning. The language in the results section has been made neutral to reflect this clarified stance.
>  3. *Parameter counts:* We have now added parameter counts for each model in the revised version (see **Table 14**, formerly Table 12, Section 4). The table caption has been expanded to specify the total trainable parameters for each configuration.
>
> **Optional Revisions**
> We have updated **Fig. 5** to include a difference plot between two randomly sampled real subsets (10k each), providing a baseline for assessing the significance of real–fake spectral differences. Additionally, we have replaced the ROC curves in **Figures 7, 8, and 9** with DET curves using the standard scikit-learn implementation, improving readability and aligning the evaluation with common ASV and spoofing analysis practices.
>
> **Possible Future Directions**
> We have incorporated them into the Limitations and Future Directions (**section 6**) of the revised manuscript, including notes on creating speaker-disjoint test splits, and exploring lighter-weight sequence backbones such as Moment, TimesFM, and Granite TSPulse. These additions help frame clear and meaningful next steps for improving both the dataset and the SAFARI-LLM architecture.